# GeoAda: Efficiently Finetune Geometric Diffusion Models with Equivariant Adapters

**Wanjia Zhao**, **Jiaqi Han**, **Siyi Gu, Mingjian Jiang, James Zou, Stefano Ermon**
Department of Computer Science
Stanford University

## Abstract

Geometric diffusion models have shown remarkable success in molecular dynamics and structure generation. However, efficiently fine-tuning them for downstream tasks with varying geometric controls remains underexplored. In this work, we propose an SE(3)-equivariant adapter framework (GeoAda) that enables flexible and parameter-efficient fine-tuning for controlled generative tasks without modifying the original model architecture. GeoAda introduces a structured adapter design: control signals are first encoded through coupling operators, then processed by a trainable copy of selected pretrained model layers, and finally projected back via decoupling operators followed by an equivariant zero-initialized convolution. By fine-tuning only these lightweight adapter modules, GeoAda preserves the model's geometric consistency while mitigating overfitting and catastrophic forgetting. We theoretically prove that the proposed adapters maintain SE(3)-equivariance, ensuring that the geometric inductive biases of the pretrained diffusion model remain intact during adaptation. We demonstrate the wide applicability of GeoAda across diverse geometric control types, including frame control, global control, subgraph control, and a broad range of application domains such as particle dynamics, molecular dynamics, human motion prediction, and molecule generation. Empirical results show that GeoAda achieves state-of-the-art fine-tuning performance while preserving original task accuracy, whereas other baselines experience significant performance degradation due to overfitting and catastrophic forgetting.

## 1 Introduction

Diffusion models have emerged as powerful generative frameworks across a wide range of domains, including image synthesis [44, 38], robotics [36, 27], and molecular generation [23, 41, 39, 42]. In particular, geometric diffusion models [11] which incorporate spatial and symmetry-aware inductive biases have shown strong empirical performance in tasks such as particle dynamic prediction [15, 17, 28], molecular generation [3, 14] and protein-ligand binding structure prediction [8]. By modeling data in an equivariant network, these models are able to capture complex geometric relationships essential for physical and chemical systems.

However, despite their strong task-specific performance, existing geometric diffusion models lack the ability to generalize across tasks. In particular, it remains unclear how a model pretrained on one geometric generation task can be effectively adapted to a new task involving additional or different control signals. This limitation is especially pronounced in real-world molecular applications, where the available data across tasks are often highly imbalanced, and collecting labeled pretraining data for every new condition is costly and time-consuming. Without a mechanism for transfer, models must be retrained from scratch for each new task, which is inefficient and often leads to overfitting or loss of previously learned capabilities.

---

*Equal contribution. Correspondence to wanjiazh@cs.stanford.edu. Code is available here.

39th Conference on Neural Information Processing Systems (NeurIPS 2025).

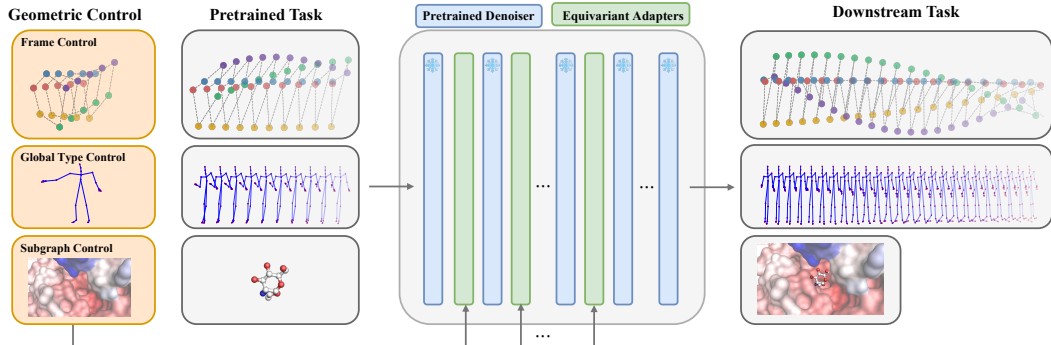

**Figure 1:** Overall framework of GeoAda. The model integrates diverse control signals, including frame, global type, and subgraph controls through lightweight equivariant adapters inserted into the frozen pretrained denoiser.

To this end, we propose a general and efficient framework(GeoAda) that enables the transfer of geometric diffusion models across diverse downstream tasks with minimal computational overhead. Inspired by the success of ControlNet [44] in conditional image generation, we introduce an equivariant adapter module that augments a pretrained geometric diffusion model with task-specific control capability. The Equivariant Adapter comprises two key components: 1) The equivariant adapter block that operates through a structured sequence—where control signals are encoded via coupling operators, processed by a trainable copy of selected pretrained model layers, and then decoded via decoupling operators. 2) Equivariant zero convolution, which acts as a safeguard for the original score by zeroing out the conditional contribution at initialization without blocking gradient updates. This design preserves the model's SE(3)-equivariance and allows modular, task-specific adaptation without altering the original model architecture. In addition to being lightweight and flexible, the adapter is parameter-efficient and implicitly regularized, thereby mitigating overfitting and preserving the performance of the pretrained model.

In summary, we make the following contributions: **1.** We propose an equivariant adapter framework (GeoAda) for geometric diffusion models that enables efficient task adaptation with minimal overhead. The adapter modules are lightweight and operate as plug-and-play components, allowing flexible conditioning on new control signals without architectural modifications to the pretrained model. **2.** GeoAda is parameter-efficient, introducing minimal overhead for downstream tasks compared to full fine-tuning, which updates the entire model and incurs substantial memory and computational costs. **3.** By freezing the pretrained model and introducing trainable adapters, GeoAda imposes implicit regularization, helping to mitigates overfitting and avoids catastrophic forgetting, thereby preserving performance on the pretraining task. **4.** We carefully design the adapter architecture to be SE(3)-equivariant, ensuring that the adapted model retains the geometric inductive bias and the theoretical benefits of equivariant diffusion models, including SE(3)-invariant marginal distributions during generation. **5.**We evaluate GeoAda across diverse geometric control types, including Frame Fontrol, Global Type Control, Subgraph Control, and a wide range of application domains, such as particle dynamics, molecular dynamics, human motion prediction, and molecule generation. GeoAda consistently matches or outperforms full fine-tuning baselines on downstream task, while avoiding performance degradation on the original pretrain task—a common failure mode of naive tuning and prompt-base approaches.

## 2 Related Work

**Geometric diffusion models.** Recent diffusion models have been extended to 3D geometric data, with SE(3) equivariance enabling physically consistent generation for tasks like molecular design and trajectory modeling. One of the earliest efforts, EDM [14] introduced an SE(3)-equivariant framework for 3D molecule generation that significantly improved sample quality. GeoDiff [42] pioneered this by learning stable molecular conformations through SE(3)-invariant diffusion, while GeoLDM [41, 39] advanced scalability and controllability via structured latent spaces. GCDM [23] advanced large molecule generation by incorporating geometry-complete local frames and chirality-sensitive features into SE(3)-equivariant networks. TargetDiff [9] further extended these models to structure-based

drug design by generating molecules conditioned on protein targets through an SE(3)-equivariant processor. Beyond molecular applications, diffusion augmented with geometric inductive bias has been explored in other domains such as 3D shape and scene generation [2] and robotics [36, 27]. Beyond static geometric modeling, GeoTDM [11] and EquiJump [5] address dynamic 3D systems by introducing temporal attention mechanisms. However, existing geometric diffusion models lack cross-task generalization. Our framework enables efficient adaptation to new controls.

**Finetuning for (geometric) graphs.** Finetuning for geometric GNNs generally falls into two categories: prompt-based and adapter-based methods. Pioneering prompt-based approaches [34, 20] introduce virtual class-prototype nodes with learnable links for edge prediction pre-trained models, but lack generalizability to alternative pre-training strategies. Meanwhile, works like GPF [6] explore universal prompt-based tuning by adding shared learnable vectors to all node features in the graph. Adapter-based methods, exemplified by AdapterGNN [18], insert lightweight modules into GNN layers, achieving parameter-efficient adaptation across diverse graph domains.

**Finetuning diffusion models.** Recent research has proposed various strategies for fine-tuning diffusion models with improved efficiency, control [44], and alignment [37]. ELEGANT [35] formulates fine-tuning as an entropy-regularized control problem, directly optimizing entropy-enhanced rewards with neural SDEs. ControlNet [44] improves controllability by adding lightweight trainable branches to frozen diffusion backbones. Prompt Diffusion [38] enables training-free in-context learning for image-to-image tasks via example-based conditioning. However, fine-tuning diffusion models in geometric domains (e.g., particles, molecules) remains underexplored. GeoAda addresses this gap by enabling efficient and effective adaptation of diffusion models to geometric tasks.

## 3 Preliminaries

**Geometric graphs and trajectories.** We represent a *geometric graph* as $\mathcal{G} = (\mathcal{V}, \mathcal{E})$ where $\mathcal{V}$ is the set of nodes and $\mathcal{E}$ is the set of edges. In particular, each node $i$ is equipped with certain node feature $\mathbf{h}_i \in \mathbb{R}^H$ representing its type or physical property, and the Euclidean coordinate $\mathbf{x}_i \in \mathbb{R}^3$ representing its spatial position. An edge exists between node $i$ and $j$ if they bear certain connectivity through, *e.g.*, chemical bonds, or spatial proximity with a distance smaller than a cutoff. A *trajectory* is a generalization of geometric graph in the dynamical setting where the coordinates $\mathbf{x}_i^{[T]} \in \mathbb{R}^{3 \times T}$ are augmented with an additional temporal dimension, where $T$ is the number of frames.

**Geometric diffusion models.** Geometric diffusion models are a family of generative models for capturing the distribution of geometric graphs and/or trajectories. Given an input data point $\mathcal{G}_0$, they feature a forward noising process that gradually perturbs the clean data with a transition $q(\mathcal{G}_\tau | \mathcal{G}_0)$ where $\mathcal{G}_T$ converges to a tractable prior. A neural network $\epsilon_\theta(\mathcal{G}_\tau, \tau)$ (*a.k.a.* the denoiser) is learned to approximate the Stein score [33] through denoising score matching [31, 32, 13], which will be leveraged to derive the transition kernel $p_\theta(\mathcal{G}_{\tau-1} | \mathcal{G}_\tau)$ in the reverse process at sampling time. Notably, a core distinction of *geometric* diffusion models from others is that they enforce an SE(3)-invariant marginal, *i.e.*,

$$p_\theta(\mathcal{G}_0) = p_\theta(g \cdot \mathcal{G}_0), \qquad \forall g \in \text{SE}(3), \tag{1}$$

by parameterizing the denoiser $\epsilon_\theta$ with an SE(3)-equivariant architecture [14, 42], *i.e.*,

$$\epsilon_\theta(g \cdot \mathcal{G}_\tau, \tau) = g \cdot \epsilon_\theta(\mathcal{G}_\tau, \tau), \tag{2}$$

where SE(3) is the Special Euclidean group consisting of all rotations and translations in 3D.

## 4 Method

In this section, we detail our approach, equivariant adapter for geometric diffusion models. We first specify three types of controls in § 4.1 that are ubiquitously enforced to geometric diffusion models in various downstream tasks. In § 4.2, we propose an architecture-agnostic and principled recipe for encoding such controls that seamlessly enables finetuning on the pretrained denoiser. In § 4.3, we present our design of GeoAda, a plug-in-and-play adapter module tuned for each downstream task that unlocks transferability.

## 4.1 Geometric Controls for Geometric Diffusion Models

In this work, we aim to transfer the generation capability of pretrained geometric diffusion models to downstream tasks where additional *geometric controls* present. Specifically, we are concerned with three different types of geometric controls, namely global type control $\mathbb{C}_G$, subgraph control $\mathbb{C}_S$, and frame control $\mathbb{C}_F$, as detailed below.

**Global type control.** Each global type control $\tilde{\mathbf{c}} \in \mathbb{C}_G$ is a vector in $\mathbb{R}^K$ describing certain global signal enforced on the geometric graph, such as an encoding of the class label, some quantum chemical property of the molecule [14], or even the embedding of some text prompt [22].

**Subgraph control.** Each subgraph control $\tilde{\mathcal{G}} = (\tilde{\mathcal{V}}, \tilde{\mathcal{E}}) \in \mathbb{C}_S$ is represented as a geometric graph with the set of nodes $\tilde{\mathcal{V}}$ and edges $\tilde{\mathcal{E}}$. Subgraph control widely exists in scenarios where generating a geometric graph conditioned on another fixed subgraph is of interest. For example, in the task of pocket-conditioned ligand generation [8], the protein pocket is viewed as the fixed subgraph $\tilde{\mathcal{G}}$ while a geometric diffusion model is learned to generate the ligand, conditioned on $\tilde{\mathcal{G}}$.

**Frame control.** Each frame control in $\mathbb{C}_F$ takes the form of a sequence of additional $\tilde{T}$ frames, namely $\tilde{\mathbf{x}}_i^{[\tilde{T}]} \in \mathbb{R}^{3 \times \tilde{T}}$ for each node $i$. Frame controls are enforced in cases when, *e.g.*, a trajectory has been partially observed and the model is expected to generate the future or missing frames conditioned on the observed frames.

## 4.2 Encoding Geometric Controls

In this subsection, we propose a simple yet effective approach for incorporating the geometric controls into the denoiser *without* modifying its architecture. Such feature is critical since it enables us to initialize the denoiser fully with the pretrained checkpoint when performing finetuning on downstream tasks, thus significantly alleviating optimization overheads and potential inconsistencies in the parameter space. More importantly, our design is also guaranteed to preserve the equivariance of the denoiser, a fundamental principle that leads to the success of geometric diffusion models.

In form, given the denoiser $\epsilon_\theta(\mathcal{G}_\tau, \tau)$, we seek to devise $\epsilon_\theta(\mathcal{G}_\tau, \tau, \mathcal{C})$ where $\mathcal{C} \in \mathbb{C}_G \cup \mathbb{C}_S \cup \mathbb{C}_F$ is any of the control we specified in § 4.1. Our core observation lies in that each type of the control can be encoded through certain coupling operator $\mathbf{f}(\mathcal{G}_\tau, \mathcal{C})$ of the input noised graph $\mathcal{G}_\tau$ and control $\mathcal{C}$, and a corresponding decoupling operator $\mathbf{g}$ that extracts the scores on the nodes and frames in $\mathcal{G}_\tau$ from the output of $\epsilon_\theta$. We introduce our design of $\mathbf{f}$ and $\mathbf{g}$ with respect to different controls as follows.

**Global type control.** For global type control $\mathcal{C}_G := \tilde{\mathbf{c}} \in \mathbb{R}^K$, we design $\mathbf{f}$ as a node-wise addition of the input node feature and a linear transformation of the control $\tilde{\mathbf{c}}$, *i.e.*, $\mathcal{V}', \mathcal{E}' = \mathbf{f}(\mathcal{V}, \mathcal{E}, \mathcal{C})$, where

$$\mathcal{V}' = (\{\mathbf{x}_i\}, \{\mathbf{h}_i + \sigma(\tilde{\mathbf{c}})\}), \qquad \mathcal{E}' = \mathcal{E}, \tag{3}$$

where $\sigma : \mathbb{R}^K \mapsto \mathbb{R}^H$ is an MLP that lifts the control signal to the node feature space. We use identity function as the decoupling operator $\mathbf{g}$.

**Subgraph control.** For subgraph control $\mathcal{C}_S := \tilde{\mathcal{G}}$, the coupling operator $\mathbf{f}$ is realized by computing the supergraph of the input $\mathcal{G}$ and the control $\tilde{\mathcal{G}}$, *i.e.*, $\mathcal{V}', \mathcal{E}' = \mathbf{f}(\mathcal{V}, \mathcal{E}, \mathcal{C})$, where

$$\mathcal{V}' = \mathcal{V} \cup \tilde{\mathcal{V}}, \qquad \mathcal{E}' = \mathcal{E} \cup \tilde{\mathcal{E}}. \tag{4}$$

The decoupling operator $\mathbf{g}$ is implemented by extracting the features of subgraph that corresponds to the nodes in the input graph $\mathcal{G}$ from the output of $\epsilon_\theta$.

**Frame control.** For frame control $\mathcal{C}_F := \{\tilde{\mathbf{x}}_i^{[\tilde{T}]}\}$, we implement $\mathbf{f}$ as a concatenation of the input frames and the frame control, *i.e.*, $\mathcal{V}', \mathcal{E}' = \mathbf{f}(\mathcal{V}, \mathcal{E}, \mathcal{C})$, where

$$\mathcal{V}' = \left(\{\text{concat}(\mathbf{x}_i^{[T]}, \tilde{\mathbf{x}}_i^{[\tilde{T}]})\}, \{\mathbf{h}_i\}\right), \qquad \mathcal{E}' = \mathcal{E}, \tag{5}$$

and $\mathbf{g}$ performs the reverse operation by discarding the frames corresponding to $[\tilde{T}]$ from the output of $\epsilon_\theta$.

**Proposition 4.1** (Equivariance of control encoding). *If the denoiser $\epsilon_\theta$ is SE(3)-equivariant, the composition $\mathbf{g} \circ \epsilon_\theta \circ \mathbf{f}$ is also SE(3)-equivariant, for all controls $\mathcal{C} \in \mathbb{C}$.*

## 4.3 Equivariant Adapters

With the control encoding in § 4.2, a straightforward approach to leverage a pretrained diffusion model on downstream tasks is to perform supervised finetuning (SFT). However, SFT usually induces suboptimal empirical performance, since **1.** SFT is parameter-inefficient since each gradient update is conducted on all parameters of the pretrained model; **2.** the full-parameter finetuning is prone to overfitting with limited amount of finetuning data; and **3.** the model finetuned after SFT loses performance guarantee on the original task, a phenomenon widely acknowledged as catastrophic forgetting.

To address these challenges, we draw inspiration from the successful application of adapters on image diffusion models, *e.g.*, ControlNet [45], to devise a diffusion adapter for geometric diffusion models. Our approach, dubbed equivariant adapter, is a lightweight tunable module plugged-in on top of the pretrained model, which is optimized for each downstream task.

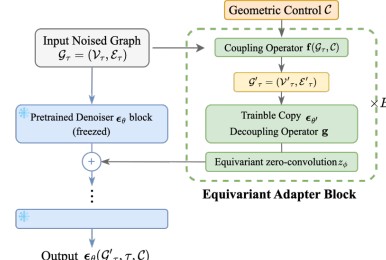

**Figure 2:** Overall framework of GeoAda. A control signal $\mathcal{C}$ is injected into the noised graph $\mathcal{G}_\tau$ and processed by an equivariant adapter block. The adapter output is added to the frozen denoiser and repeated $B$ times to produce the final output.

**The equivariant adapter block.** In detail, each equivariant adapter block is responsible for processing the control signal and fusing it into the score produced by the pretrained model, whose parameters are always frozen at finetuning stage. Each adapter block consists of, in a sequential manner, the coupling operator $\mathbf{f}$, a *trainable copy* of the corresponding layers in pretrained model $\boldsymbol{\epsilon}_{\theta'}$, the decoupling operator $\mathbf{g}$, followed by an equivariant *zero-convolution* layer.

Specifically, the composition $\mathbf{g} \circ \boldsymbol{\epsilon}_{\theta'} \circ \mathbf{f}$, as depicted in § 4.2, functions altogether as a conditional score network $\boldsymbol{\epsilon}_{\theta'}(\mathcal{G}_\tau, \tau, \mathcal{C})$ that captures the bias of the control signal on the original score $\boldsymbol{\epsilon}_\theta(\mathcal{G}_\tau, \tau)$ while ensuring the SE(3)-equivariance of the conditional score. Moreover, $\boldsymbol{\epsilon}_{\theta'}$ can be initialized as a subset of the layers in the pretrained model $\boldsymbol{\epsilon}_\theta$, thus reducing the total number of tunable parameters compared with SFT. While the selection strategy can be arbitrary, empirically we have found that selecting the first layer for every $K$ consecutive layers from the pretrained model performs more favorably compared with naive choices such as the initial or last several layers, under the same parameter budget (*c.f.*, § 5.4).

**Equivariant zero-convolution.** While the equivariant adapter block offers an parameter-efficient way of modeling the conditional score, its non-zero initialization introduces additional noise when it is added to the original score $\boldsymbol{\epsilon}_\theta$, leading to instability at the beginning of the finetuning stage. To alleviate such issue, we borrow insight from the *zero-convolution* module proposed in [44] that acts as a safeguard of the original score by zeroing out the conditional score at initialization without blocking the gradient update.

For any $(\{\mathbf{x}_i\}, \{\mathbf{h}_i\})$, equivariant zero-convolution is given by

$$z_\phi(\{\mathbf{x}_i\}, \{\mathbf{h}_i\}) = (\{\phi_\mathbf{x} \cdot (\mathbf{x}_i - \bar{\mathbf{x}})\}, \{\phi_\mathbf{h} \odot \mathbf{h}_i\}),  \qquad (6)$$

where $\bar{\mathbf{x}} = \frac{1}{N} \sum_{i=1}^N \mathbf{x}_i$ is the center-of-mass of the input graph, and $\phi_\mathbf{x} \in \mathbb{R}, \phi_\mathbf{h} \in \mathbb{R}^H$ are learnable parameters initialized all as zero. By such design, we guarantee that each equivariant adapter block, when equipped with equivariant zero-convolution, yields a rotation-equivariant and translation-invariant output, hence the SE(3)-equivariance of the conditional score after adding the output to the original score. Furthermore, the output of equivariant adapter block will always be zero at initialization, which does not affect the original score, thus enabling smooth and noiseless optimization when tuning the adapter.

## 4.4 Overall Framework

The overall framework of our adapter is depicted in Fig. 2. In general, our adapter is comprised of $B$ equivariant adapter blocks, where each block is a sequential stack of the coupling operator $\mathbf{f}$, the trainable copy of one layer of the denoiser, the decoupling operator $\mathbf{g}$, and a zero-convolution module. At finetuning stage, all parameters in the original denoiser are frozen while the trainable copies and coefficients in zero-convolution are updated through gradient coming from minimizing the denoising loss

$$\mathcal{L}_{\text{finetune}}(\theta', \phi) = \mathbb{E}_{\boldsymbol{\epsilon} \sim \mathcal{N}(\mathbf{0}, \mathbf{I}), (\mathcal{G}, \mathcal{C}) \sim \mathbb{D}, \tau \in \text{Unif}(0, T)} \|\boldsymbol{\epsilon}_\theta(\mathcal{G}_\tau, \tau) + \mathbf{s}_{\theta', \phi}(\mathcal{G}_\tau, \tau, \mathcal{C}) - \boldsymbol{\epsilon}\|_2^2, \qquad (7)$$

where $\mathbb{D}$ is the downstream dataset, $\mathcal{G}_\tau$ is the noised graph drawn from $q(\mathcal{G}_\tau|\mathcal{G}_0)$, and $\mathbf{s}_{\theta',\phi}$ refers to our proposed equivariant diffusion adapter. At inference time, we use $\epsilon_\theta(\mathcal{G}_\tau, \tau) + \mathbf{s}_{\theta',\phi}(\mathcal{G}_\tau, \tau, \mathcal{C})$ as the conditional score when computing the reverse transition kernel $p(\mathcal{G}_{\tau-1}|\mathcal{G}_\tau, \mathcal{C})$.

Our GeoAda offers several key advantages over standard supervised fine-tuning (SFT):**1.** The adapter modules are lightweight and operate as plug-and-play components, allowing flexible conditioning on new control signals without architectural modifications to the pretrained model. **2.** Parameter-efficient, as only a subset of trainable adapter modules are introduced. **3.** By freezing the Pretrained model and only optimizing lightweight adapters, the method imposes implicit regularization, helping to prevent overfitting. **4.** Through the careful design of SE(3)-equivariant adapter blocks and zero convolutions, GeoAda guarantees equivariance throughout the tuning process, thereby retaining the theoretical benefits of geometric diffusion models.

## 5 Experiment

We evaluate GeoAda across three categories of additional fine-tuning controls: (1) Frame control during dynamic prediction (§ 5.1), (2) Global type control in human motion prediction(§ 5.2), and (3) Subgraph control in molecule generation (§ 5.3). We also performed ablation studies on core design choices and present some observations in §5.4.

**Baselines.** We compare with three types of baselines: (1) *Fine-tuning* methods, including **Full FT** , which fine-tunes the entire model, and **PARTIAL-**$k$ [12, 16, 45], which updates only the last $k$ layers of the pre-trained model; (2) *Prompt-based* methods, including **GPF**, **GPF-plus** [6], which both inject learnable prompt features into the input space. And **Prompt-Template** maps new inputs to pre-training-style inputs using manually designed graph templates, specifically for the conditional case. (3) *Head-only* tuning methods, where **MLP-**$k$ freezes the pre-trained model and uses a $k$-layer MLP as the prediction head. To preserve equivariance, we replace the MLP with an EGTN block in our implementation. More details can be found in App. 9.2.

**Implementation.** The input data are processed as geometric graphs. For both trajectory and global control settings, we follow the same experimental setup as GeoTDM [11], adopting EGTN as the backbone model with three GeoAda blocks and a hidden dimension of 128. For subgraph control, the base model follows the configuration of TargetDiff [8]. We use $\mathcal{T} = 1000$ and linear noise schedule [13]. More details in App. 9.1.

### 5.1 Frame Control

**Task setup.** For pre-training, we use the first 10 frames as the condition and train the model to predict the trajectory over the following 20 frames. In the downstream task, we adopt a different setup where the model observes 15 conditional frames and predicts the next 20 frames. We evaluate all models on both the original task and the new task to assess their generalization and adaptability across different settings.

**Metrics.** For conditional trajectory generation, we employ Average Discrepancy Error (ADE) and Final Discrepancy Error (FDE), which are widely adopted for trajectory forecasting [43, 40], given by $\text{ADE}(\mathbf{x}^{[T]}, \mathbf{y}^{[T]}) = \frac{1}{TN} \sum_{t=0}^{T-1} \sum_{i=0}^{N-1} \|\mathbf{x}_i^{(t)} - \mathbf{y}_i^{(t)}\|_2$, and $\text{FDE}(\mathbf{x}^{[T]}, \mathbf{y}^{[T]}) = \frac{1}{N} \sum_{i=0}^{N-1} \|\mathbf{x}_i^{(T-1)} - \mathbf{y}_i^{(T-1)}\|_2$. For probabilistic models, we report average ADE and FDE derived from $K = 5$ samples. For unconditional trajectory generation, we report three complementary scores: The Marginal score measures statistical alignment by computing the mean absolute error (MAE) between binned distributions of model-generated and ground-truth coordinates (or bond lengths for MD17). The Classification score is the cross-entropy of a binary classifier trained to distinguish generated trajectories from real ones, offering insight into sample realism. The Prediction score measures the mean squared error (MSE) of a sequence model trained on generated data and tested on real trajectories, reflecting the utility of generated samples for downstream prediction. For more detailed metric definitions, please refer to Appendix 9.5.

### 5.1.1 Particle Dynamic

**Experimental Setup.** We adopt the CHARGED PARTICLES dataset [17, 28] for particle dynamics simulation. In this dataset, $N = 5$ particles with randomly assigned charges of either $+1$ or $-1$ interact via Coulomb forces, resulting in complex, non-linear trajectories. We use 3000 trajectories for training, 2000 for validation, and 2000 for testing. We explore two settings: (1) Conditional trajectory generation: we use the first 10 frames of each trajectory as input to predict the subsequent 20 frames during pretraining, and 15 frames as input during finetuning to predict the next 20 frames. (2) Unconditional trajectory generation: we generate trajectories of length 20 from scratch during pretraining. During finetuning, we condition on the first 10 frames and predict the next 20 frames.

**Table 1:** Comparisons on CHARGED PARTICLES dataset.(all results reported by $\times 10^{-1}$).($\uparrow$) / ($\downarrow$) denotes whether a larger / smaller number is preferred. "NaN" denotes generation collapse due to numerical instability, typically observed in baseline models after fine-tuning on original task. "–"indicates that the baseline Prompt-Tem requires explicit conditioning and cannot be applied when no conditioning frame is given.

| Setting | Uncondition | | | | | Condition | | | |
|---|---|---|---|---|---|---|---|---|---|
| Task | Downstream | | Pretrain | | | Downstream | | Pretrain | |
| Metric | ADE($\downarrow$) | FDE($\downarrow$) | Marg($\downarrow$) | Class($\uparrow$) | Pred($\downarrow$) | ADE($\downarrow$) | FDE($\downarrow$) | ADE($\downarrow$) | FDE($\downarrow$) |
| Pretrain | nan | nan | $\underline{0.079}_{\pm0.000}$ | $\mathbf{5.149}_{\pm0.285}$ | $\mathbf{0.109}_{\pm0.004}$ | 11.826$_{\pm0.133}$ | 20.395$_{\pm0.249}$ | $\underline{1.177}_{\pm0.018}$ | $\underline{2.815}_{\pm0.037}$ |
| Full FT | $\mathbf{1.093}_{\pm0.014}$ | $\underline{2.676}_{\pm0.024}$ | 1.025$_{\pm0.000}$ | nan | nan | $\underline{1.106}_{\pm0.007}$ | $\mathbf{2.590}_{\pm0.040}$ | 5.998$_{\pm0.041}$ | 11.75$_{\pm0.107}$ |
| Prompt-Tem | - | - | - | - | - | 1.723$_{\pm0.014}$ | 3.703$_{\pm0.061}$ | nan | nan |
| PARTIAL-$k$ [12] | 1.685$_{\pm0.006}$ | 3.594$_{\pm0.040}$ | 1.016$_{\pm0.000}$ | nan | nan | 1.409$_{\pm0.009}$ | 3.330$_{\pm0.042}$ | 9.325$_{\pm0.064}$ | 11.94$_{\pm0.149}$ |
| MLP-$k$ | 6.258$_{\pm0.947}$ | 9.111$_{\pm2.628}$ | 1.015$_{\pm0.000}$ | 0.00$_{\pm0.000}$ | 5.740$_{\pm2.914}$ | 1.503$_{\pm0.016}$ | 3.338$_{\pm0.039}$ | 3924 | 3950 |
| GPF [6] | 1.643$_{\pm0.014}$ | 3.671$_{\pm0.029}$ | 1.027$_{\pm0.000}$ | nan | nan | 1.575$_{\pm0.017}$ | 3.390$_{\pm0.050}$ | nan | nan |
| GPF-plus [6] | 1.596$_{\pm0.011}$ | 3.574$_{\pm0.024}$ | 1.023$_{\pm0.000}$ | nan | nan | 1.648$_{\pm0.009}$ | 3.670$_{\pm0.030}$ | nan | nan |
| **GeoAda** | $\underline{1.119}_{\pm0.019}$ | $\mathbf{2.669}_{\pm0.022}$ | $\mathbf{0.079}_{\pm0.000}$ | $\underline{5.134}_{\pm0.247}$ | $\underline{0.111}_{\pm0.006}$ | $\mathbf{1.105}_{\pm0.012}$ | $\underline{2.621}_{\pm0.033}$ | $\mathbf{1.175}_{\pm0.033}$ | $\mathbf{2.806}_{\pm0.033}$ |

**Results.** We present the results in Table 1, with the following observations. Under both the unconditional and conditional trajectory generation settings, GeoAda achieves comparable or better performance than Full FT on the downstream task (conditioning on the first 10 frames to predict the next 20), while only tuning half the number of parameters. Furthermore, it consistently outperforms other fine-tuning and prompt-based baselines, achieving 35.69% improvement on ADE and 21.29% on FDE. On the original pretraining task, all baselines exhibit substantial performance degradation, with some failing to generate valid diffusion samples, indicating that these methods suffer from overfitting and catastrophic forgetting due to excessive adaptation to the downstream task."In contrast, by leveraging equivariant zero convolutions, GeoAda retains the pretrained model's performance.

### 5.1.2 Molecular Dynamics

**Experimental setup.** We employ the MD17 [3] dataset, which contains the DFT-simulated molecular dynamics trajectories of 8 small molecules, with the number of atoms for each molecule ranging from 9 (Ethanol and Malonaldehyde) to 21 (Aspirin). For each molecule, 5000 trajectories are used for training and 1000/1000 for validation and testing, uniformly sampled along the time dimension. Different from [40], we explicitly involve the hydrogen atoms which contribute most to the vibrations of the trajectory, leading to a more challenging task. The node feature is the one-hot encodings of atomic number [29] and edges are connected between atoms within three hops measured in atomic bonds [30].

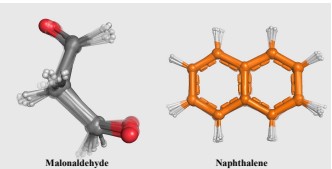

**Figure 3:** Visualization results of GeoAda on Malonaldehyde and Naphthalene from MD17 dataset.

**Results.** As shown in Table 2, GeoAda achieves state-of-the-art performance across all five molecular systems in the MD17 dataset, indicating strong transferability in geometric diffusion models. In downstream fine-tuning task, the method consistently matches or exceeds the performance of full fine-tuning and outperforms other prior methods by an average of 18.94% in ADE and 18.22% in FDE. Importantly, when returning to the original pretraining task, it retains performance comparable to the pretrained model, while both fine-tuning and prompt-based methods exhibit significant degradation or collapse due to overfitting. More experiment results on Malonaldehyde and Naphthalene in App. 10.1.

**Table 2:** Comparisons for Molecular Dynamics prediction on MD17 dataset (all results reported by $\times 10^{-1}$). The best results are highlighted in bold. Results averaged over 5 runs. "NaN" denotes generation collapse due to numerical instability, typically observed in baseline models after fine-tuning on the original task.

| Scenarios | Aspirin | | | | Benzene | | | | Ethanol | | | |
|---|---|---|---|---|---|---|---|---|---|---|---|---|
| Task | Downstream | | Pretrain | | Downstream | | Pretrain | | FT | | Pretrain | |
| Metric | ADE | FDE | ADE | FDE | ADE | FDE | ADE | FDE | ADE | FDE | ADE | FDE |
| Pretrain | 3.782±0.010 | 7.345±0.016 | 1.062±0.002 | 1.857±0.013 | 0.603±0.000 | 1.325±0.008 | 0.241±0.000 | 0.393±0.002 | 3.263±0.010 | 4.357±0.014 | 0.999±0.009 | 1.856±0.032 |
| Full FT | 0.929±0.002 | 1.602±0.005 | 1.323±0.003 | 2.280±0.016 | 0.217±0.001 | 0.360±0.002 | nan | nan | 0.997±0.007 | 1.906±0.002 | inf | inf |
| PARTIAL-$k$ [12] | 1.071±0.003 | 1.875±0.009 | 1.439±0.003 | 2.407±0.007 | 0.249±0.005 | 0.407±0.003 | 0.318±0.110/nan | 0.491±0.001/nan | 1.288±0.008 | 2.161±0.018 | 6.502±4.574 | 4.636±2.423 |
| MLP-$k$ | 1.132±0.004 | 1.921±0.004 | 1.579±0.005 | 2.641±0.007 | 0.248±0.012 | 0.412±0.004 | nan | nan | 1.301±0.001 | 2.275±0.021 | 2.228±0.004 | 2.716±0.022 |
| Prompt-Tem | 1.197±0.008 | 2.014±0.030 | 1.571±0.010 | 2.668±0.024 | 0.241±0.012 | 0.409±0.021 | nan | nan | 1.207±0.018 | 2.194±0.081 | nan | nan |
| GPF [6] | 1.130±0.006 | 1.909±0.024 | 3.260±0.015/inf | 4.272±0.025/inf | 0.246±0.001 | 0.415±0.004 | nan | nan | 1.239±0.023 | 2.233±0.059 | 2.420±0.156 | 3.519±0.056 |
| GPF-plus [6] | 1.014±0.006 | 1.962±0.021 | 2.243±0.009 | 3.349±0.018 | 0.234±0.010 | 0.331±0.057 | 1.118±0.006 | 1.083±0.010 | 1.137±0.025 | 2.048±0.046 | 3.240±0.081/inf | 4.074±0.171/inf |
| **GeoAda** | **0.891**±0.003 | **1.533**±0.008 | **1.060**±0.003 | **1.852**±0.012 | **0.191**±0.000 | **0.319**±0.002 | **0.240**±0.002 | **0.394**±0.005 | **0.905**±0.007 | **1.745**±0.010 | **0.995**±0.005 | 1.867±0.019 |

## 5.2 Global Type Control

**Experimental setup.** The CMU Mocap dataset is a commonly used dataset for human pose prediction, which includes 8 action categories. A single pose has 38 body joints in the original dataset, among which we choose 25 joints following the configuration of MSR-GCN [4], using 10 frames as input to predict the subsequent 25 frames. For pre-training, we construct a dataset by combining the three most frequent actions: directing traffic, washing windows, and giving basketball signals. The remaining five actions are used as the downstram task fine-tuning dataset.

**Results** We report short-term and long-term motion prediction results on the CMU Mocap dataset in Tables 3 and 4. More results on jumping and soccer scenarios are in App. 10.2. GeoAda consistently achieves state-of-the-art performance across all action categories and time horizons. In this setting, the pretraining dataset is significantly larger than the downstream task dataset (30k vs. 245–1345 datapoints). As a result, naïve fine-tuning and prompt-based methods are highly prone to overfitting to the limited downstream training data, leading to notably degraded performance. Moreover, they are more likely to fail to generate valid samples on the original

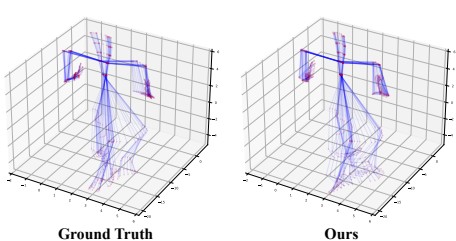

**Ground Truth**      **Ours**

**Figure 4:** Visualization of Running trajectory

pretraining task, indicating a severe loss of pretrained knowledge and catastrophic forgetting. In contrast, GeoAda benefits from the implicit regularization effect of the adapter, which mitigates overfitting and preserves the performance of the pretrained model.

**Table 3:** Comparisons for short-term prediction on 5 action categories of the CMU Mocap dataset. The best results are highlighted in bold. Results averaged over 5 runs (std in App. 10.2).

| scenarios | running | | | | pretrain | | | | walking | | | | pretrain | | | | basketball | | | | pretrain | | | |
|---|---|---|---|---|---|---|---|---|---|---|---|---|---|---|---|---|---|---|---|---|---|---|---|---|
| millisecond (ms) | 80 | 160 | 320 | 400 | 80 | 160 | 320 | 400 | 80 | 160 | 320 | 400 | 80 | 160 | 320 | 400 | 80 | 160 | 320 | 400 | 80 | 160 | 320 | 400 |
| Pretrain | 28.74 | 57.99 | 126.20 | 159.37 | 7.941 | 16.84 | 39.91 | 52.45 | 15.49 | 30.66 | 68.71 | 89.23 | 7.941 | 16.84 | 39.91 | 52.45 | 17.92 | 35.89 | 78.48 | 101.12 | 7.941 | 16.84 | 39.91 | 52.45 |
| Full FT | 20.34 | 35.26 | 60.58 | 70.20 | nan | nan | nan | nan | 10.01 | 15.12 | 23.98 | 28.49 | nan | nan | nan | nan | 16.95 | 30.33 | 58.28 | 71.93 | nan | nan | nan | nan |
| PARTIAL-$k$ [12] | 20.47 | 36.04 | 68.15 | 82.59 | nan | nan | nan | nan | 10.47 | 16.97 | 30.97 | 37.69 | nan | nan | nan | nan | 17.54 | 32.03 | 62.96 | 78.95 | nan | nan | nan | nan |
| MLP-$k$ | 23.35 | 44.44 | 84.27 | 101.42 | nan | nan | nan | nan | 10.86 | 18.74 | 35.78 | 44.86 | nan | nan | nan | nan | 17.60 | 34.76 | 72.90 | 86.34 | nan | nan | nan | nan |
| GPF [6] | 18.48 | 31.94 | 58.80 | 73.56 | 21.38 | 35.44 | 65.23 | 79.55 | 10.79 | 16.79 | 28.32 | 33.48 | 22.04 | 37.10 | 71.24 | 88.23 | 18.48 | 31.94 | 58.80 | 73.56 | 21.38 | 35.44 | 65.23 | 79.55 |
| GPF-plus [6] | 19.11 | 31.93 | 49.85 | 56.67 | nan | nan | nan | nan | 10.17 | 16.23 | 26.29 | 31.54 | nan | nan | nan | nan | 17.17 | 31.86 | 58.48 | 72.71 | nan | nan | nan | nan |
| **GeoAda** | 18.70 | 33.56 | 50.26 | 55.54 | 7.972 | 16.91 | 40.06 | 52.61 | 8.92 | 13.82 | 22.99 | 26.68 | 7.932 | 16.90 | 38.96 | 52.54 | 16.85 | 29.71 | 57.59 | 71.19 | 7.898 | 16.79 | 39.70 | 52.50 |

**Table 4:** Comparisons for long-term prediction on 5 action categories of the CMU Mocap dataset. The best results are highlighted in bold. Results averaged over 5 runs (std in App. 10.2).

| scenarios | running | | pretrain | | walking | | pretrain | | basketball | | pretrain | |
|---|---|---|---|---|---|---|---|---|---|---|---|---|
| millisecond (ms) | 560 | 1000 | 560 | 1000 | 560 | 1000 | 560 | 1000 | 560 | 1000 | 560 | 1000 |
| Pretrain | 219.16 | 314.85 | **77.06** | 130.51 | 129.43 | 212.94 | **77.06** | 130.51 | 143.49 | 223.99 | 77.06 | 130.51 |
| Full FT | 85.14 | 97.02 | nan | nan | 36.92 | 52.58 | nan | nan | 94.59 | 132.34 | nan | nan |
| PARTIAL-$k$ [12] | 102.85 | 108.47 | nan | nan | 51.36 | 84.72 | 118.82 | 182.88 | 106.84 | 146.27 | nan | nan |
| MLP-$k$ | 127.67 | 131.59 | nan | nan | 62.97 | 102.34 | nan | nan | 107.30 | 149.58 | nan | nan |
| GPF [6] | 61.92 | 71.42 | nan | nan | 42.37 | 52.24 | 119.43 | 171.74 | 97.16 | 128.29 | nan | nan |
| GPF-plus [6] | 63.56 | 71.60 | nan | nan | 41.31 | 56.47 | nan | nan | 104.54 | 130.76 | 104.51 | 155.02 |
| **GeoAda** | **60.88** | **70.22** | 77.22 | 130.17 | **34.52** | **50.49** | 78.12 | **129.97** | **91.03** | **120.35** | 76.94 | **129.81** |

## 5.3 Subgraph Control

**Experimental setup.** We adopt the QM9 [26] and GEOM-Drugs [1] dataset for pretraining a model for molecule generation, use the CrossDocked2020 dataset [7] for finetuning protein-ligand pair generation. QM9 [26] contains 130k small molecules with atom types (H, C, N, O, F). GEOM-Drugs [1]

**Table 5:** Summary of binding affinity and molecular properties of reference molecules and molecules generated by GeoAda and baselines. (↑) / (↓) denotes whether a larger / smaller number is preferred.

| Methods | Vina Score (↓) | | Vina Min (↓) | | Vina Dock (↓) | | High Affinity(↑) | | QED(↑) | | SA(↑) | | Diversity(↑) | |
|---|---|---|---|---|---|---|---|---|---|---|---|---|---|---|
| | Avg. | Med. | Avg. | Med. | Avg. | Med. | Avg. | Med. | Avg. | Med. | Avg. | Med. | Avg. | Med. |
| liGAN [25] | - | - | - | - | -6.33 | -6.20 | 21.1% | 11.1% | 0.39 | 0.39 | 0.59 | 0.57 | 0.66 | 0.67 |
| GraphBP [19] | - | - | - | - | -4.80 | -4.70 | 14.2% | 6.7% | 0.43 | 0.45 | 0.49 | 0.48 | **0.79** | **0.78** |
| AR [21] | -5.75 | -5.64 | -6.18 | -5.88 | -6.75 | -6.62 | 37.9% | 31.0% | 0.51 | 0.50 | 0.63 | 0.63 | 0.70 | 0.70 |
| Pocket2Mol [24] | -5.14 | -4.70 | -6.42 | -5.82 | -7.15 | -6.79 | 48.4% | 51.0% | **0.56** | **0.57** | **0.74** | **0.75** | 0.69 | 0.71 |
| TargetDiff [10] | -5.47 | -6.30 | -6.64 | **-6.83** | **-7.80** | **-7.91** | 58.1% | 59.1% | 0.48 | 0.48 | 0.58 | 0.58 | 0.72 | 0.71 |
| **GeoAda (qm9)** | **-5.54** | **-6.31** | -6.64 | -6.46 | -7.62 | -7.64 | 57.4% | 58.2% | 0.49 | 0.51 | 0.58 | 0.58 | 0.74 | 0.75 |
| **GeoAda (Geom)** | -5.54 | -6.01 | **-6.68** | -6.32 | -7.64 | -7.71 | **58.3%** | **59.3%** | 0.48 | 0.50 | 0.58 | 0.58 | 0.76 | 0.75 |
| Reference | -6.36 | -6.41 | -6.71 | -6.49 | -7.45 | -7.26 | - | - | 0.48 | 0.47 | 0.73 | 0.74 | - | - |

**Table 6:** Jensen-Shannon divergence of bond distance distributions between reference and generated molecules. (↓)

| Bond | liGAN | AR | Pocket2Mol | TargetDiff | GeoAda (qm9) | GeoAda (Geom) |
|---|---|---|---|---|---|---|
| C−C | 0.601 | 0.609 | 0.496 | 0.369 | **0.243** | 0.269 |
| C=C | 0.665 | 0.620 | 0.561 | 0.505 | **0.377** | 0.393 |
| C−N | 0.634 | 0.474 | 0.416 | 0.363 | **0.363** | 0.396 |
| C=N | 0.749 | 0.635 | 0.629 | 0.550 | 0.300 | **0.299** |
| C−O | 0.656 | 0.492 | 0.454 | 0.421 | 0.418 | 0.428 |
| C=O | 0.661 | 0.558 | 0.516 | 0.461 | 0.279 | **0.257** |
| C:C | 0.497 | 0.451 | 0.416 | **0.263** | 0.305 | 0.335 |
| C:N | 0.638 | 0.552 | 0.487 | **0.235** | 0.297 | 0.330 |

**Table 7:** Percentage of different ring sizes for reference and model generated molecules.

| Ring Size | Ref. | liGAN | AR | Pocket2Mol | TargetDiff | GeoAda (qm9) | GeoAda (geom) |
|---|---|---|---|---|---|---|---|
| 3 | 1.7% | 28.1% | 29.9% | 0.1% | 0.0% | 0.0% | 0.0% |
| 4 | 0.0% | 15.7% | 0.0% | 0.0% | 2.8% | 6.7% | 5.8% |
| 5 | 30.2% | 29.8% | 16.0% | 16.4% | 30.8% | 47.2% | 45.8% |
| 6 | 67.4% | 22.7% | 51.2% | 80.4% | 50.7% | 69.1% | 78.2% |
| 7 | 0.7% | 2.6% | 1.7% | 2.6% | 12.1% | 23.5 | 21.3% |
| 8 | 0.0% | 0.8% | 0.7% | 0.3% | 2.7% | 5.3% | 4.7% |
| 9 | 0.0% | 0.3% | 0.5% | 0.1% | 0.9% | 3.8% | 1.6% |

is a large-scale dataset of 430k molecular conformers with heavy atoms, and we keep the lowest energy conformation for each molecule. Following the common setup for CrossDocked2020 [10], we obtain 100k complexes for training and 100 novel complexes for testing. Since CrossDocked2020 has different atom type configuration from QM9 and GEOM-Drugs, we limit the atom type to (H, C, N, O, F, P, S, Cl). Following [10], proteins and ligands are expressed as 3D atom coordinates and one-hot vectors containing the atom types.

**Implementation.** Following prior work [10], we use the Adam optimizer with a learning rate of 0.001 and $\beta$ values of (0.95, 0.999). Batch size is set to 4 and gradient clipping set to 8. To balance the atom type and position losses, we scale the atom type loss by a factor of $\lambda = 100$.

**Results.** We evaluate molecular properties and molecular structures of the proposed model and baselines on target-aware molecule generation in Table 5, 6, and 7. Baseline models are trained on CrossDocked2020 under explicit protein conditioning. GeoAda matches, and in multiple cases surpasses the strongest baselines on all metrics, generating ligand molecules that maintain realistic structures, high binding affinity, comparable drug-likeness and sythetic accessibility. The lightweight adapter can inject subgraph (pocket) control into a broadly pretrained geometric diffusion model, achieving or surpassing task-specific baselines that rely on end-to-end training with protein context.

### 5.4 Ablations and Observations

**Observation of the sudden convergence phenomenon** Similar to the phenomenon observed in ControlNet [44], we also observe a *sudden convergence phenomenon* in our training process. As shown in Figure 5, between step 4500 and 4700, both training loss and validation MSE drop abruptly rather than gradually. To investigate this behavior, we conducted inference using the saved checkpoints from steps 3600 and 5600, and observed a notable performance jump between steps 4400 and 4800, which corresponds to significant reductions in ADE and FDE by 68.3% and 73.4%.

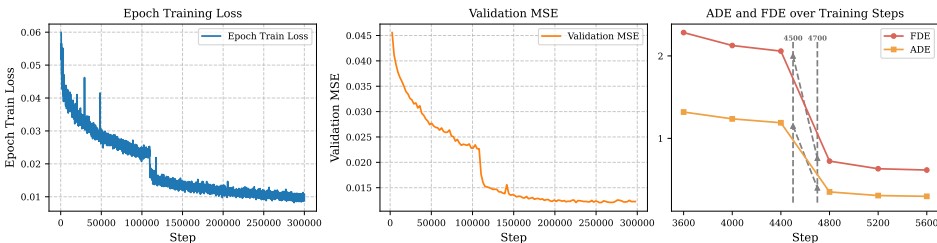

**Figure 5:** The sudden convergence phenomenon

**Parameter efficiency analysis** As shown in Appendix 10.3.1, we explore the impact of varying the number of equivariant zero layers. Increasing the number of trainable copy layers generally improves

performance, but introduces more parameters and computational cost, revealing a trade-off between performance and efficiency. We also reported the number of tunable parameters for different tuning strategies in Table 21. Except for full fine-tuning, which is substantially larger, all other methods, including GeoAda, use comparable parameter sizes.

**Ablation on Equivariant Zero Convolutions**    We evaluate two variants to assess the role of equivariant zero convolution: (1) replacing it with Gaussian-initialized standard convolutions, and (2) replacing each trainable copy with a single convolution layer (see App. 10.3.2). Both modifications result in notable performance drops, underscoring the importance of zero initialization and structural design for stable and effective fine-tuning.

# 6  Conclusion

We present GeoAda, a parameter-efficient and SE(3)-equivariant adapter framework for geometric diffusion models. It enables effective adaptation to diverse geometric control tasks without modifying the pretrained backbone, preserving both performance and geometric consistency.

# 7  Acknlowlegdements

This project was funded in part by ARO (W911NF-21-1-0125), ONR (N00014-23-1-2159), and the CZ Biohub.

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

# Appendix

## 8 Proof

Below is the explanation and proof of Proposition 4.1:

*Proof.* Since $\boldsymbol{\epsilon}_\theta$ is SE(3)-equivariant by assumption, we have for any $h \in$ SE(3),

$$\boldsymbol{\epsilon}_\theta(h \cdot \mathcal{G}') = h \cdot \boldsymbol{\epsilon}_\theta(\mathcal{G}'), \quad \text{where } \mathcal{G}' = \mathbf{f}(\mathcal{G}_\tau, \mathcal{C}).$$

We consider each component:

- The coupling operator $\mathbf{f}$ augments $\mathcal{G}_\tau$ with control $\mathcal{C}$ in a way that respects the SE(3) structure: global controls modify features invariantly; subgraph controls are merged geometrically; frame controls concatenate along the temporal axis. Thus, $\mathbf{f}$ is SE(3)-equivariant.

- The decoupling operator $\mathbf{g}$ selects a subset of nodes or frames without altering their coordinates. Therefore, it commutes with SE(3) action: $\mathbf{g}(h \cdot \mathcal{G}'') = h \cdot \mathbf{g}(\mathcal{G}'')$.

Combining the above, we explicitly see that for any $h \in$ SE(3) defined as $h(\mathbf{x}) = R\mathbf{x} + \mathbf{d}$, we have:

$$\begin{aligned}
&\mathbf{g} \circ \boldsymbol{\epsilon}_\theta \circ \mathbf{f}(h \cdot (\mathcal{G}_\tau, \mathcal{C})) \\
&= \mathbf{g}\left(\boldsymbol{\epsilon}_\theta\left(h \cdot \mathbf{f}(\mathcal{G}_\tau, \mathcal{C})\right)\right) \\
&= \mathbf{g}\left(h \cdot \boldsymbol{\epsilon}_\theta\left(\mathbf{f}(\mathcal{G}_\tau, \mathcal{C})\right)\right) \\
&= h \cdot \mathbf{g}\left(\boldsymbol{\epsilon}_\theta\left(\mathbf{f}(\mathcal{G}_\tau, \mathcal{C})\right)\right) \\
&= R \cdot (\mathbf{g} \circ \boldsymbol{\epsilon}_\theta \circ \mathbf{f}(\mathcal{G}_\tau, \mathcal{C})) + \mathbf{d} \\
&= h \cdot (\mathbf{g} \circ \boldsymbol{\epsilon}_\theta \circ \mathbf{f}(\mathcal{G}_\tau, \mathcal{C}))
\end{aligned}$$

Therefore, the composed function $\mathbf{g} \circ \boldsymbol{\epsilon}_\theta \circ \mathbf{f}$ is SE(3)-equivariant.

$\square$

## 9 More Details on Experiments

### 9.1 Hyper-parameters

We provide the detailed hyper-parameters of GeoAda in Table 8. We adopt Adam optimizer with betas $(0.9, 0.999)$ and $\epsilon = 10^{-8}$. For all experiments, we use the linear noise schedule per [13] with $\beta_{\text{start}} = 0.02$ and $\beta_{\text{end}} = 0.0001$.

**Table 8:** Hyper-parameters of GeoAda in the experiments.

|           | n_layer | hidden | time_emb_dim | $\mathcal{T}$ | batch_size | learning_rate |
|-----------|---------|--------|--------------|---------------|------------|---------------|
| N-body    | 6       | 128    | 32           | 1000          | 128        | 0.0001        |
| MD        | 6       | 128    | 32           | 1000          | 128        | 0.0001        |
| CMU Mocap | 6       | 64     | 32           | 100           | 128        | 0.0001        |

### 9.2 Baselines

**Full FT.**

**Full FT** fully fine-tunes the pre-trained model $f$ during downstream training. The entire model is updated to fit the target task.

**PARTIAL-$k$.**

**PARTIAL-$k$** fine-tunes only the last $k$ layers of the model $f$, while freezing the remaining layers. This method balances adaptability with parameter efficiency by limiting the number of updated layers.

**Graph Prompt Feature (GPF).**

In **GPF**, the pre-trained encoder $f$ is kept frozen, and a learnable prompt vector $p$ is injected into the input feature space. During training, only the prompt vector $p$ and the prediction head $\theta$ are optimized. This method enables task adaptation through a lightweight, task-specific prompt without modifying the backbone model. In our implementation, we replace the original MLP head with a three-layer Equivariant Geometric Trajectory Network (EGTN), which ensures the projection head maintains geometric consistency with the model.

**Graph Prompt Feature-Plus (GPF-plus).**

Extending GPF, this variant constructs node-wise prompt vectors using $k$ learnable basis vectors $p_1^b, \ldots, p_k^b$ and a set of learnable linear weights $a_1, \ldots, a_k$. These components are used to compute node-specific prompts $p_i$ via a compositional mechanism. The model $f$ remains frozen, while prediction head $\theta$, learnable basis vectors $p_i^b$, andlearnable linear weights $a_i$ are optimized.

**Prompt-Template**

We prepend a learnable prompt layer(Equivariant Geometric Trajectory Network) to adapt new inputs to the distribution seen during pretraining, following with prediction head $\theta$.

**MLP-$k$ (EGTN-$k$).**

This baseline freezes the entire pre-trained model $f$ and replaces the prediction head with a $k$-layer multilayer perceptron (MLP). To preserve equivariance in our setting, we replace the MLP with an Equivariant Geometric Trajectory Network (EGTN) block. Only the EGTN-based head is trained during the downstream task.

## 9.3 Details of the datasets

### 9.3.1 Global Type Control

**Pretrain Dataset**    The statistics of the pretrained datasets on Global Type Control are presented in Table. 9.

Table 9: Pretrain Dataset statistics by Global Type.

| Type | Washwindow | Directing Traffic | Basketball Signal | Pretrain |
|------|-----------|-------------------|-------------------|----------|
| train | 12126 | 9557 | 7776 | 29459 |
| val | 1342 | 2346 | 1920 | 5588 |
| test | 1342 | 2346 | 1920 | 5588 |

**Downstream datasets**    The statistics of the downstream datasets utilized for the models pretrained on Global Type Control are presented in Table. 10.

Table 10: Downstream Dataset statistics by Global Type.

| Type | Running | Walking | Jumping | Basketball | Soccer |
|------|---------|---------|---------|------------|--------|
| train | 245 | 869 | 1345 | 1044 | 1210 |
| val | 47 | 145 | 1008 | 254 | 264 |
| test | 47 | 145 | 1008 | 254 | 264 |

## 9.4 Model

### 9.4.1 Geometric Trajectory Diffusion Models

**Unconditional Generation**    For unconditional generation, we model the trajectory distribution subject to SE(3)-invariance. The following theorem provides constraints for the prior and transition kernel.

**Theorem 9.1.** *If the prior $p_{\mathcal{T}}(\mathbf{x}_{\mathcal{T}}^{[T]})$ is SE(3)-invariant, and the transition kernels $p_{\tau-1}(\mathbf{x}_{\tau-1}^{[T]} \mid \mathbf{x}_{\tau}^{[T]}), \forall \tau \in \{1, \cdots, \mathcal{T}\}$ are SE(3)-equivariant, then the marginal $p_{\tau}(\mathbf{x}_{\tau}^{[T]})$ at any step $\tau \in \{0, \cdots, \mathcal{T}\}$ is also SE(3)-invariant.*

**Prior in the translation-invariant subspace.** The prior is built on a translation-invariant subspace $\mathcal{X}_{\mathbf{P}} \subset \mathcal{X}$, induced by a linear transformation $\mathbf{P}$:

$$\mathbf{P} = \mathbf{I}_D \otimes \left( \mathbf{I}_{TN} - \frac{1}{TN} \mathbf{1}_{TN} \mathbf{1}_{TN}^{\top} \right)$$

which results in a restricted Gaussian distribution supported only on the subspace, denoted $\tilde{\mathcal{N}}(\mathbf{0}, \mathbf{I})$, and is isotropic and SO(3)-invariant. To sample, one samples from $\mathcal{N}(\mathbf{0}, \mathbf{I})$ and projects it onto the subspace.

**Transition kernel.** The transition kernel is parameterized in the subspace $\mathcal{X}_{\mathbf{P}}$, given by:

$$p_{\boldsymbol{\theta}}(\tilde{\mathbf{x}}_{\tau-1}^{[T]} \mid \tilde{\mathbf{x}}_{\tau}^{[T]}) = \tilde{\mathcal{N}}(\tilde{\boldsymbol{\mu}}_{\boldsymbol{\theta}}(\tilde{\mathbf{x}}_{\tau}^{[T]}, \tau), \sigma_{\tau}^2 \mathbf{I})$$

where the mean function $\tilde{\boldsymbol{\mu}}_{\boldsymbol{\theta}}$ is SO(3)-equivariant. The function is re-parameterized as:

$$\tilde{\boldsymbol{\mu}}_{\boldsymbol{\theta}}(\tilde{\mathbf{x}}_{\tau}^{[T]}, \tau) = \frac{1}{\sqrt{\alpha_{\tau}}} \left( \tilde{\mathbf{x}}_{\tau}^{[T]} - \frac{\beta_{\tau}}{\sqrt{1 - \bar{\alpha}_{\tau}}} \tilde{\boldsymbol{\epsilon}}_{\boldsymbol{\theta}}(\mathbf{x}_{\tau}^{[T]}, \tau) \right)$$

where $\tilde{\boldsymbol{\epsilon}}_{\boldsymbol{\theta}} = P \circ \mathbf{f}_{\boldsymbol{\theta}}$ is an SO(3)-equivariant adaptation of the proposed EGTN.

**Training and inference.** The VLB is optimized for training, with the objective:

$$\mathcal{L}_{\text{uncond}} := \mathbb{E}_{\mathbf{x}_0^{[T]}, \tilde{\boldsymbol{\epsilon}} \sim \tilde{\mathcal{N}}(\mathbf{0}, \mathbf{I}), \tau \sim \text{Unif}(1, \mathcal{T})} \left[ \| \tilde{\boldsymbol{\epsilon}} - \tilde{\boldsymbol{\epsilon}}_{\boldsymbol{\theta}}(\tilde{\mathbf{x}}_{\tau}^{[T]}, \tau) \|^2 \right]$$

The inference process involves projecting intermediate samples onto the subspace $\mathcal{X}_{\mathbf{P}}$.

**Conditional Generation** In conditional generation, the target distribution is SE(3)-equivariant with respect to the given frames. The following theorem provides constraints for the prior and transition kernel.

**Theorem 9.2.** *If the prior $p_{\mathcal{T}}(\mathbf{x}_{\mathcal{T}}^{[T]} \mid \mathbf{x}_c^{[T_c]})$ is SE(3)-equivariant, and the transition kernels $p_{\tau-1}(\mathbf{x}_{\tau-1}^{[T]} \mid \mathbf{x}_{\tau}^{[T]}, \mathbf{x}_c^{[T_c]}), \forall \tau \in \{1, \cdots, \mathcal{T}\}$ are SE(3)-equivariant, the marginal $p_{\tau}(\mathbf{x}_{\tau}^{[T]} \mid \mathbf{x}_c^{[T_c]})$, $\forall \tau \in \{0, \cdots, \mathcal{T}\}$ is SE(3)-equivariant.*

**Flexible equivariant prior.** We provide a guideline for distinguishing feasible prior designs. The prior $\mathcal{N}(\boldsymbol{\mu}(\mathbf{x}_c^{[T_c]}), \mathbf{I})$ is SE(3)-equivariant if $\boldsymbol{\mu}(\mathbf{x}_c^{[T_c]})$ is SE(3)-equivariant. The mean function $\boldsymbol{\mu}(\mathbf{x}_c^{[T_c]})$ serves as an anchor to transition geometric information from the given frames to the target distribution. For instance, the anchor can be defined as:

$$\mathbf{x}_r^{[T]} = \mathbf{1}_{T \times N} \otimes \sum_{s \in [T_c]} w^{(s)} \bar{\mathbf{x}}_c^{(s)}$$

where the weights satisfy $\sum_{s \in [T_c]} w^{(s)} = 1$.

The weights $\mathbf{w}^{(t,s)}$ are derived as:

$$\mathbf{W}_{t,s} = [\boldsymbol{\gamma} \otimes \hat{\mathbf{h}}_c^{[T_c]}]_{t,s} \in \mathbb{R}^N, \quad \mathbf{w}^{(t,s)} = \begin{cases} \mathbf{W}_{t,s} & \text{if } s < T_c - 1, \\ \mathbf{1}_N - \sum_{s=0}^{T_c-2} \mathbf{W}_{t,s} & \text{if } s = T_c - 1. \end{cases}$$

where $\boldsymbol{\gamma} \in \mathbb{R}^T$ are learnable parameters, ensuring the constraint for translation equivariance.

**Transition kernel.** To match the proposed prior, we modify both the forward and reverse processes. The forward process is defined as:

$$q(\mathbf{x}_{\tau}^{[T]} \mid \mathbf{x}_{\tau-1}^{[T]}, \mathbf{x}_c^{[T_c]}) := \mathcal{N}(\mathbf{x}_{\tau}^{[T]}; \mathbf{x}_r + \sqrt{1 - \beta_{\tau}}(\mathbf{x}_{\tau-1}^{[T]} - \mathbf{x}_r), \beta_{\tau} \mathbf{I}),$$

which ensures that $q(\mathbf{x}_{\mathcal{T}}^{[T]} \mid \mathbf{x}_c^{[T_c]})$ matches the equivariant prior $\mathbf{x}_r$ (proof in App. **??**). The reverse transition kernel is:

$$p_{\tau-1}(\mathbf{x}_{\tau-1}^{[T]} \mid \mathbf{x}_{\tau}^{[T]}, \mathbf{x}_c^{[T_c]}) = \mathcal{N}(\boldsymbol{\mu}_{\boldsymbol{\theta}}(\mathbf{x}_{\tau}^{[T]}, \tau, \mathbf{x}_c^{[T_c]}), \sigma_{\tau}^2 \mathbf{I}).$$

We adopt the noise prediction objective for the reverse process, rewriting $\boldsymbol{\mu_\theta}$ as:

$$\boldsymbol{\mu_\theta}(\mathbf{x}_\tau^{[T]}, \mathbf{x}_c^{[T_c]}, \tau) = \mathbf{x}_r^{[T]} + \frac{1}{\sqrt{\alpha_\tau}}\left(\mathbf{x}_\tau^{[T]} - \mathbf{x}_r^{[T]} - \frac{\beta_\tau}{\sqrt{1-\bar{\alpha}_\tau}}\boldsymbol{\epsilon_\theta}(\mathbf{x}_\tau^{[T]}, \mathbf{x}_c^{[T_c]}, \tau)\right),$$

where the denoising network $\boldsymbol{\epsilon_\theta}$ is implemented as an EGTN with translation invariance, ensuring the translation equivariance of $\boldsymbol{\mu_\theta}$.

**Training and inference.** Optimizing the VLB of our diffusion model leads to the following objective:

$$\mathcal{L}_{\text{cond}} := \mathbb{E}_{\mathbf{x}_0^{[T]}, \mathbf{x}_c^{[T_c]}, \boldsymbol{\epsilon}\sim\mathcal{N}(\mathbf{0},\mathbf{I}), \tau\sim\text{Unif}(1,\mathcal{T})}\left[\|\boldsymbol{\epsilon} - \boldsymbol{\epsilon_\theta}(\mathbf{x}_\tau^{[T]}, \mathbf{x}_c^{[T_c]}, \tau)\|^2\right].$$

## 9.5 Evaluation Metrics in the Unconditional Case

All these metrics are evaluated on a set of model samples with the same size as the testing set.

**Marginal score** is computed as the absolute difference of two empirical probability density functions. Practically, we collect the $x, y, z$ coordinates at each time step marginalized over all nodes in all systems in the predictions and the ground truth (testing set). Then we split the collection into 50 bins and compute the MAE in each bin, finally averaged across all time steps to obtain the score. Note that on MD17, instead of computing the pdf on coordinates, we compute the pdf on the length of the chemical bonds, which is a clearer signal that correlates to the validity of the generated MD trajectory, since during MD simulation the bond lengths are usually stable with very small vibrations. Marginal score gives a broad statistical measurement how each dimension of the generated samples align with the original data.

**Classification score** is computed as the cross-entropy loss of a sequence classification model that aims to distinguish whether the trajectory is generated by the model or from the testing set. To be specific, we construct a dataset mixed by the generated samples and the testing set, and randomly split it into 80% and 20% subsets for training and testing. Then the model is trained on the training set and the classification score is computed as the cross-entropy on the testing set. We use a 1-layer EqMotion with a classification head as the model. The classification score provided intuition on how difficult it is to distinguish the generated samples and the original data.

**Prediction score** is computed as the MSE loss of a train-on-synthetic-test-on-real sequence to sequence model. In detail, we train a 1-layer EqMotion on the sampled dataset with the task of predicting the second half of the trajectory given the first half. We then evaluate the model on the testing set and report the MSE as the prediction score. Prediction score provides intuition on the capability of the generative model on generating synthetic data that well aligns with the ground truth.

# 10 More Experiments and Discussions

## 10.1 Molecular

Additional experimental results on the Malonaldehyde and Naphthalene are shown below:

**Table 11:** Comparisons for Molecular Dynamics prediction on MD17 dataset (all results reported by $\times 10^{-1}$). The best results are highlighted in bold. Results averaged over 5 runs

| Scenarios | Malonaldehyde | | | | Naphthalene | | | |
|---|---|---|---|---|---|---|---|---|
| Task | Downstream | | Pretrain | | Downstream | | Pretrain | |
| Metric | ADE | FDE | ADE | FDE | ADE | FDE | ADE | FDE |
| **Pretrain** | $3.235_{\pm0.012}$ | $5.189_{\pm0.023}$ | $\mathbf{0.962}_{\pm0.007}$ | $1.584_{\pm0.021}$ | $1.416_{\pm0.003}$ | $2.268_{\pm0.005}$ | $0.714_{\pm0.002}$ | $0.972_{\pm0.006}$ |
| **FT** | $\underline{0.897}_{\pm0.002}$ | $\underline{1.511}_{\pm0.009}$ | $1.405_{\pm0.006}$ | $2.237_{\pm0.023}$ | $\mathbf{0.555}_{\pm0.001}$ | $\underline{0.867}_{\pm0.010}$ | nan | nan |
| **PARTIAL-$k$ [12]** | $0.981_{\pm0.004}$ | $1.675_{\pm0.015}$ | $1.230_{\pm0.003}$ | $2.110_{\pm0.006}$ | $0.653_{\pm0.002}$ | $0.903_{\pm0.003}$ | $2.083_{\pm0.009}$ | $1.629_{\pm0.007}$ |
| **MLP-$k$** | $0.997_{\pm0.005}$ | $1.694_{\pm0.010}$ | $1.291_{\pm0.004}$ | $2.051_{\pm0.015}$ | $0.718_{\pm0.001}$ | $0.969_{\pm0.005}$ | nan | nan |
| **Prompt-Tem** | $1.092_{\pm0.019}$ | $2.003_{\pm0.056}$ | $2.323_{\pm0.024}$ | $3.351_{\pm0.081}$ | $0.972_{\pm0.006}$ | $1.593_{\pm0.021}$ | nan | nan |
| **GPF [6]** | $1.176_{\pm0.012}$ | $1.931_{\pm0.030}$ | $282.1_{\pm157.9}$/inf | $24.97_{\pm24.43}$/inf | $0.758_{\pm0.002}$ | $1.005_{\pm0.005}$ | nan | nan |
| **GPF-plus [6]** | $1.018_{\pm0.003}$ | $1.793_{\pm0.010}$ | $3.527_{\pm0.501}$ | $4.719_{\pm1.397}$ | $0.717_{\pm0.003}$ | $0.873_{\pm0.006}$ | $1.891_{\pm0.056}$ | $2.674_{\pm0.156}$ |
| **GeoAda** | $\mathbf{0.862}_{\pm0.002}$ | $\mathbf{1.414}_{\pm0.014}$ | $\underline{0.963}_{\pm0.007}$ | $\mathbf{1.573}_{\pm0.018}$ | $\underline{0.581}_{\pm0.002}$ | $\mathbf{0.822}_{\pm0.004}$ | $\mathbf{0.714}_{\pm0.001}$ | $\mathbf{0.969}_{\pm0.007}$ |

## 10.2 Human Motion

Additional experimental results on the *jumping* and *soccer* scenarios are presented below. We also report the standard deviations across all experiments.

**Table 12:** Short-term prediction on **running** from the CMU Mocap dataset.

| scenarios | running | | | | pretrain | | | |
|---|---|---|---|---|---|---|---|---|
| millisecond (ms) | 80 | 160 | 320 | 400 | 80 | 160 | 320 | 400 |
| Pretrain | $28.74_{\pm0.34}$ | $57.99_{\pm0.33}$ | $126.20_{\pm0.93}$ | $159.37_{\pm1.15}$ | $\mathbf{7.941}_{\pm0.02}$ | $\mathbf{16.84}_{\pm0.04}$ | $\mathbf{39.91}_{\pm0.43}$ | $\mathbf{52.45}_{\pm0.07}$ |
| Full FT | $20.34_{\pm0.32}$ | $35.26_{\pm0.19}$ | $60.58_{\pm1.14}$ | $70.20_{\pm1.36}$ | nan | nan | nan | nan |
| PARTIAL-$k$ | $20.47_{\pm0.32}$ | $36.04_{\pm0.40}$ | $68.15_{\pm0.70}$ | $82.59_{\pm0.86}$ | nan | nan | nan | nan |
| MLP-$k$ | $23.35_{\pm0.39}$ | $44.44_{\pm0.71}$ | $84.27_{\pm1.58}$ | $101.42_{\pm2.11}$ | nan | nan | nan | nan |
| GPF | $19.17_{\pm0.28}$ | $32.85_{\pm0.66}$ | $52.83_{\pm1.03}$ | $60.90_{\pm1.54}$ | $21.38_{\pm0.97}$ | $35.44_{\pm1.05}$ | $65.23_{\pm1.76}$ | $79.55_{\pm1.11}$ |
| GPF-plus | $19.11_{\pm0.54}$ | $31.93_{\pm0.81}$ | $49.85_{\pm1.21}$ | $56.67_{\pm1.39}$ | nan | nan | nan | nan |
| **GeoAda** | $\mathbf{18.70}_{\pm0.37}$ | $\mathbf{33.56}_{\pm0.25}$ | $50.26_{\pm0.42}$ | $\mathbf{55.54}_{\pm0.36}$ | $7.972_{\pm0.02}$ | $16.91_{\pm0.05}$ | $40.06_{\pm0.42}$ | $52.61_{\pm0.07}$ |

**Table 13:** Short-term prediction on **walking** from the CMU Mocap dataset.

| scenarios | walking | | | | pretrain | | | |
|---|---|---|---|---|---|---|---|---|
| millisecond (ms) | 80 | 160 | 320 | 400 | 80 | 160 | 320 | 400 |
| Pretrain | $15.49_{\pm0.07}$ | $30.66_{\pm0.16}$ | $68.71_{\pm0.33}$ | $89.23_{\pm0.43}$ | $7.941_{\pm0.02}$ | $16.84_{\pm0.04}$ | $39.91_{\pm0.43}$ | $52.45_{\pm0.07}$ |
| Full FT | $10.01_{\pm0.17}$ | $15.12_{\pm0.14}$ | $23.98_{\pm0.13}$ | $28.49_{\pm0.19}$ | nan | nan | nan | nan |
| PARTIAL-$k$ | $10.47_{\pm0.81}$ | $16.97_{\pm0.43}$ | $30.97_{\pm0.60}$ | $37.69_{\pm0.60}$ | $17.26_{\pm0.23}$ | $31.75_{\pm0.36}$ | $66.29_{\pm0.44}$ | $84.27_{\pm0.79}$ |
| MLP-$k$ | $10.86_{\pm1.07}$ | $18.74_{\pm1.37}$ | $35.78_{\pm1.92}$ | $44.86_{\pm1.35}$ | nan | nan | nan | nan |
| GPF | $10.79_{\pm0.14}$ | $16.79_{\pm0.16}$ | $28.32_{\pm0.21}$ | $33.48_{\pm0.22}$ | $22.04_{\pm0.38}$ | $37.10_{\pm0.40}$ | $71.24_{\pm0.39}$ | $88.23_{\pm0.97}$ |
| GPF-plus | $10.17_{\pm0.16}$ | $16.23_{\pm0.12}$ | $26.29_{\pm0.22}$ | $31.54_{\pm0.19}$ | nan | nan | nan | nan |
| **GeoAda** | $\mathbf{8.92}_{\pm1.02}$ | $\mathbf{13.82}_{\pm1.26}$ | $\mathbf{22.99}_{\pm1.30}$ | $\mathbf{26.68}_{\pm1.31}$ | $7.932_{\pm0.03}$ | $16.90_{\pm0.04}$ | $38.96_{\pm0.47}$ | $52.54_{\pm0.10}$ |

**Table 14:** Short-term prediction comparison on the **basketball** action from the CMU Mocap dataset.

| scenarios | basketball | | | | pretrain | | | |
|---|---|---|---|---|---|---|---|---|
| millisecond (ms) | 80 | 160 | 320 | 400 | 80 | 160 | 320 | 400 |
| Pretrain | $17.92_{\pm0.06}$ | $35.89_{\pm0.12}$ | $78.48_{\pm0.47}$ | $101.12_{\pm0.69}$ | $7.941_{\pm0.02}$ | $16.84_{\pm0.04}$ | $39.91_{\pm0.43}$ | $\mathbf{52.45}_{\pm0.07}$ |
| FT | $16.95_{\pm0.11}$ | $30.33_{\pm0.17}$ | $58.28_{\pm0.41}$ | $71.93_{\pm0.54}$ | - | - | - | - |
| PARTIAL-$k$ | $17.54_{\pm0.12}$ | $32.03_{\pm0.43}$ | $62.96_{\pm0.86}$ | $78.95_{\pm0.98}$ | - | - | - | - |
| MLP-$k$ | $17.60_{\pm0.50}$ | $34.76_{\pm1.26}$ | $72.90_{\pm2.36}$ | $86.34_{\pm1.53}$ | - | - | - | - |
| GPF | $18.48_{\pm0.09}$ | $31.94_{\pm0.14}$ | $58.80_{\pm0.28}$ | $73.56_{\pm0.29}$ | $21.38_{\pm0.11}$ | $35.44_{\pm0.27}$ | $65.23_{\pm0.30}$ | $79.55_{\pm0.51}$ |
| GPF-plus | $17.17_{\pm0.07}$ | $31.86_{\pm0.15}$ | $58.48_{\pm0.32}$ | $72.71_{\pm0.24}$ | - | - | - | - |
| **GeoAda** | $\mathbf{16.85}_{\pm0.24}$ | $\mathbf{29.71}_{\pm0.41}$ | $\mathbf{57.59}_{\pm0.39}$ | $\mathbf{71.19}_{\pm0.48}$ | $7.898_{\pm0.05}$ | $16.79_{\pm0.05}$ | $39.70_{\pm0.04}$ | $52.50_{\pm0.07}$ |

**Table 15:** Short-term prediction comparison on the **jumping** action from the CMU Mocap dataset.

| scenarios | jumping | | | | pretrain | | | |
|---|---|---|---|---|---|---|---|---|
| millisecond (ms) | 80 | 160 | 320 | 400 | 80 | 160 | 320 | 400 |
| Pretrain | - | - | - | - | $\mathbf{7.941}_{\pm0.02}$ | $16.84_{\pm0.04}$ | $39.91_{\pm0.43}$ | $\mathbf{52.45}_{\pm0.07}$ |
| FT | - | - | - | - | $26.67_{\pm0.10}$ | $50.83_{\pm0.29}$ | $94.13_{\pm0.58}$ | $112.66_{\pm0.71}$ |
| PARTIAL-$k$ | $26.01_{\pm0.08}$ | $49.19_{\pm0.09}$ | $95.84_{\pm0.13}$ | $116.24_{\pm0.23}$ | $19.08_{\pm0.17}$ | $38.53_{\pm0.34}$ | $79.77_{\pm0.52}$ | $99.85_{\pm1.07}$ |
| MLP-$k$ | 22.63/nan | 44.68/nan | 88.93/nan | 108.43/nan | $15.32_{\pm0.30}$ | $31.92_{\pm0.26}$ | $66.50_{\pm0.42}$ | $82.55_{\pm0.93}$ |
| GPF | $28.74_{\pm0.19}$ | $51.97_{\pm0.19}$ | $97.80_{\pm0.34}$ | $117.94_{\pm0.37}$ | $_{\pm0.08}$ - | - | - | - |
| GPF-plus | - | - | - | - | - | - | - | - |
| **GeoAda** | $\mathbf{25.91}_{\pm0.09}$ | $\mathbf{48.83}_{\pm0.83}$ | $\mathbf{91.51}_{\pm0.07}$ | $\mathbf{109.24}_{\pm0.60}$ | $7.956_{\pm0.03}$ | $16.82_{\pm0.04}$ | $39.55_{\pm0.47}$ | $52.57_{\pm0.09}$ |

**Table 16:** Short-term prediction comparison on the **soccer** action from the CMU Mocap dataset.

| scenarios | soccer | | | | pretrain | | | |
|---|---|---|---|---|---|---|---|---|
| millisecond (ms) | 80 | 160 | 320 | 400 | 80 | 160 | 320 | 400 |
| Pretrain | - | - | - | - | $\mathbf{7.941}_{\pm0.02}$ | $\mathbf{16.84}_{\pm0.04}$ | $39.91_{\pm0.43}$ | $52.45_{\pm0.07}$ |
| FT | $17.65_{\pm0.17}$ | $31.43_{\pm0.35}$ | $59.76_{\pm0.44}$ | $74.30_{\pm0.54}$ | - | - | - | - |
| PARTIAL-$k$ | $18.83_{\pm0.10}$ | $32.86_{\pm0.07}$ | $64.58_{\pm0.37}$ | $81.53_{\pm0.50}$ | $14.14_{\pm0.41}$ | $24.95_{\pm0.38}$ | $50.89_{\pm0.40}$ | $64.24_{\pm0.51}$ |
| MLP-$k$ | - | - | - | - | - | - | - | - |
| GPF | $19.28_{\pm0.10}$ | $32.18_{\pm0.14}$ | $69.58_{\pm0.42}$ | $73.63_{\pm0.63}$ | $15.70_{\pm0.30}$ | $28.31_{\pm0.28}$ | $58.57_{\pm0.42}$ | $74.28_{\pm0.61}$ |
| GPF-plus | $19.11_{\pm0.18}$ | $32.03_{\pm0.31}$ | $59.67_{\pm0.46}$ | $74.25_{\pm0.65}$ | $15.22_{\pm0.23}$ | $26.22_{\pm0.29}$ | $52.02_{\pm0.47}$ | $65.17_{\pm0.53}$ |
| **GeoAda** | $\mathbf{17.04}_{\pm0.14}$ | $\mathbf{30.03}_{\pm0.12}$ | $\mathbf{53.51}_{\pm0.25}$ | $\mathbf{64.78}_{\pm0.42}$ | $7.961_{\pm0.02}$ | $16.90_{\pm0.03}$ | $39.46_{\pm0.41}$ | $52.43_{\pm0.09}$ |

**Table 17:** Long-term prediction on CMU Mocap: **Running** and **Walking**.

| scenarios | running | | pretrain | | walking | | pretrain | |
|---|---|---|---|---|---|---|---|---|
| millisecond (ms) | 560 | 1000 | 560 | 1000 | 560 | 1000 | 560 | 1000 |
| Pretrain | $219.16_{\pm2.18}$ | $314.85_{\pm3.03}$ | $\mathbf{77.06}_{\pm0.47}$ | $130.51_{\pm0.27}$ | $129.43_{\pm0.77}$ | $212.94_{\pm1.90}$ | $\mathbf{77.06}_{\pm0.47}$ | $130.51_{\pm0.27}$ |
| Full FT | $85.14_{\pm2.36}$ | $97.02_{\pm1.45}$ | nan | nan | $36.92_{\pm1.37}$ | $52.58_{\pm0.62}$ | nan | nan |
| PARTIAL-$k$ [12] | $102.85_{\pm1.68}$ | $108.47_{\pm2.03}$ | nan | nan | $51.36_{\pm1.93}$ | $84.72_{\pm0.67}$ | $118.82_{\pm0.58}$ | $182.88_{\pm0.79}$ |
| MLP-$k$ | $127.67_{\pm2.46}$ | $131.59_{\pm2.90}$ | nan | nan | $62.97_{\pm1.45}$ | $102.34_{\pm0.86}$ | nan | nan |
| GPF [6] | $61.92_{\pm1.02}$ | $71.42_{\pm1.27}$ | nan | nan | $42.37_{\pm0.31}$ | $52.24_{\pm0.38}$ | $119.43_{\pm0.60}$ | $171.74_{\pm0.55}$ |
| GPF-plus [6] | $63.56_{\pm1.54}$ | $71.60_{\pm0.95}$ | nan | nan | $41.31_{\pm0.35}$ | $56.47_{\pm0.4}$ | nan | nan |
| **GeoAda** | $\mathbf{60.88}_{\pm0.82}$ | $\mathbf{70.22}_{\pm2.02}$ | $77.22_{\pm0.45}$ | $\mathbf{130.17}_{\pm0.26}$ | $\mathbf{34.52}_{\pm2.26}$ | $\mathbf{50.49}_{\pm0.33}$ | $78.12_{\pm0.49}$ | $\mathbf{129.97}_{\pm0.30}$ |

**Table 18:** Long-term prediction on CMU Mocap: **Jumping** and **Soccer**.

| scenarios | jumping | | pretrain | | soccer | | pretrain | |
|---|---|---|---|---|---|---|---|---|
| millisecond (ms) | 560 | 1000 | 560 | 1000 | 560 | 1000 | 560 | 1000 |
| Pretrain | – | – | $77.06_{\pm0.47}$ | $130.51_{\pm0.27}$ | – | – | $77.06_{\pm0.47}$ | $130.51_{\pm0.27}$ |
| Full FT | – | – | $145.76_{\pm1.23}$ | $199.34_{\pm2.01}$ | $101.44_{\pm0.51}$ | $157.11_{\pm0.96}$ | – | – |
| PARTIAL-$k$ [12] | $149.06_{\pm0.37}$ | $181.52_{\pm0.81}$ | $135.25_{\pm0.92}$ | $186.16_{\pm1.58}$ | $113.88_{\pm0.75}$ | $170.50_{\pm1.98}$ | $89.63_{\pm1.62}$ | $142.41_{\pm1.84}$ |
| MLP-$k$ | 140.11/nan | 184.77/nan | $111.71_{\pm0.74}$ | $158.55_{\pm1.37}$ | – | – | – | – |
| GPF [6] | $151.50_{\pm0.65}$ | $194.94_{\pm0.34}$ | – | – | $100.23_{\pm0.88}$ | $151.84_{\pm1.23}$ | $104.03_{\pm0.81}$ | $161.56_{\pm1.31}$ |
| GPF-plus [6] | – | – | – | – | $101.15_{\pm0.86}$ | $153.43_{\pm0.79}$ | $90.12_{\pm1.10}$ | $142.30_{\pm1.12}$ |
| **GeoAda** | $139.46_{\pm0.29}$ | $184.01_{\pm0.60}$ | $\mathbf{76.98}_{\pm0.51}$ | $\underline{130.72}_{\pm0.26}$ | $\mathbf{84.91}_{\pm0.46}$ | $\mathbf{125.91}_{\pm0.75}$ | $\underline{77.19}_{\pm0.48}$ | $\mathbf{130.06}_{\pm0.31}$ |

**Table 19:** Long-term prediction on CMU Mocap: **Basketball**.

| scenarios | basketball | | pretrain | |
|---|---|---|---|---|
| millisecond (ms) | 560 | 1000 | 560 | 1000 |
| Pretrain | $143.49_{\pm1.19}$ | $223.99_{\pm2.23}$ | $\underline{77.06}_{\pm0.47}$ | $\underline{130.51}_{\pm0.27}$ |
| Full FT | $\underline{94.59}_{\pm0.58}$ | $132.34_{\pm1.30}$ | nan | nan |
| PARTIAL-$k$ [12] | $106.84_{\pm1.00}$ | $146.27_{\pm1.24}$ | nan | nan |
| MLP-$k$ | $107.30_{\pm2.76}$ | $149.58_{\pm2.24}$ | nan | nan |
| GPF [6] | $97.16_{\pm0.35}$ | $128.29_{\pm0.43}$ | nan | nan |
| GPF-plus [6] | $104.54_{\pm0.26}$ | $\underline{130.76}_{\pm1.28}$ | $104.51_{\pm0.57}$ | $155.02_{\pm0.51}$ |
| **GeoAda** | $\mathbf{91.03}_{\pm0.33}$ | $\mathbf{120.35}_{\pm0.44}$ | $\mathbf{76.94}_{\pm0.45}$ | $\mathbf{129.81}_{\pm0.30}$ |

## 10.3 Ablations

### 10.3.1 Parameter efficiency analysis

As shown in Table 20, we explore the impact of varying the number of equivariant adapter blocks. Increasing the number of trainable copy layers generally improves performance, but introduces more parameters and computational cost, revealing a trade-off between performance and efficiency. We have computed the number of tunable parameters for all baselines and GeoAda. The statistics are presented in Table 21. Except for Full Fine-Tuning, all methods have a comparable number of tunable parameters, ensuring a fair comparison.

**Table 20:** Different numbers of adapter blocks

| Number | ADE | FDE |
|---|---|---|
| *1* | 1.321 | 3.088 |
| *2* | 1.291 | 2.968 |
| *3* | **1.108** | **2.621** |
| *4* | 1.104 | 2.588 |
| *5* | 1.106 | 2.686 |

**Table 21:** The number of tunable parameters for different tuning strategies.

| Dataset | Tuning Strategy | Total Parameters | Tunable Parameters |
|---|---|---|---|
| Charged Particle | Full FT | 1418252 ~5.41 MB | 1418252 ~5.41 MB |
| | PARTIAL-$k$ | 1418252 ~5.41 MB | 711302 ~2.71 MB |
| | MLP-$k$ | 2125202 ~8.11 MB | 711302 ~2.71 MB |
| | Prompt-Tem | 1418252 ~5.41 MB | 711302 ~2.71 MB |
| | GPF | 2125266 ~8.11 MB | 711366 ~2.71 MB |
| | GPF-plus | 2125847 ~8.11 MB | 711947 ~2.72 MB |
| | **GeoAda** | 2125691 ~8.11 MB | 711791 ~2.72 MB |
| MD17 | Full FT | 1424268 ~5.43 MB | 1424268 ~5.43 MB |
| | PARTIAL-$k$ | 1424268 ~5.43 MB | 716166 ~2.73 MB |
| | MLP-$k$ | 2132370 ~8.13 MB | 716166 ~2.73 MB |
| | Prompt-Tem | 1424268 ~5.43 MB | 716166 ~2.73 MB |
| | GPF | 2132434 ~8.13 MB | 716230 ~2.73 MB |
| | GPF-plus | 2135079 ~8.14 MB | 718875 ~2.74 MB |
| | **GeoAda** | 2132844 ~8.14 MB | 716640 ~2.73 MB |
| CMU Mocap | Full FT | 368012 ~1.40 MB | 368012~1.40 MB |
| | PARTIAL-$k$ | 368012 ~1.40 MB | 185990~0.71 MB |
| | MLP-$k$ | 550034 ~2.10 MB | 185990~0.71 MB |
| | GPF | 550098 ~2.10 MB | 186054 ~0.71 MB |
| | GPF-plus | 553259 ~2.11 MB | 189215 ~0.72 MB |
| | **GeoAda** | 550075 ~2.10 MB | 186031~0.71 MB |

### 10.3.2 Ablative Architectures

We study the following ablative architectures as shown in Figure 6, Figure 7, Figure 8:

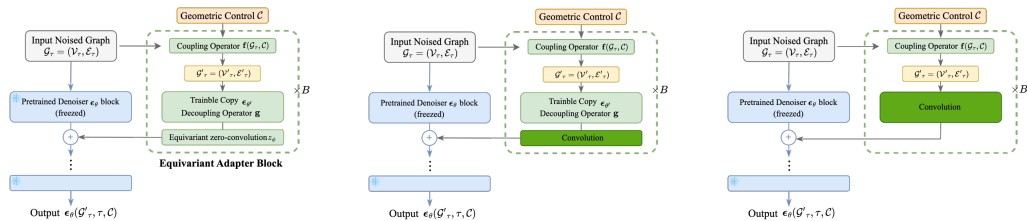

**Figure 6:** GeoAda      **Figure 7:** w/o zero convolution      **Figure 8:** w/o trainable copy

**Proposed GeoAda.** The proposed architecture in the main paper.

**Without Zero Convolution.** Replacing the zero convolutions with standard convolution layers initialized with Gaussian weights.

**Lightweight Layers.** This architecture does not use a trainable copy, and directly initializes single convolution layers.

**Results** We present the results of this ablative study in Table 22, Table 23, Table 24 and Table 25.

**Table 22:** Comparisons for Molecular Dynamics prediction on MD17 dataset (all results reported by $\times 10^{-1}$). The best results are highlighted in bold. Results averaged over 5 runs

| Scenarios | Aspirin | | | | Benzene | | | |
|---|---|---|---|---|---|---|---|---|
| Task | Downstream | | Pretrain | | Downstream | | Pretrain | |
| Metric | ADE | FDE | ADE | FDE | ADE | FDE | ADE | FDE |
| w/o trainable copy | 2.232 $\pm 0.008$ | 3.501 $\pm 0.022$ | 2.197 $\pm 0.007$ | 3.449 $\pm 0.010$ | 0.607 $\pm 0.002$ | 0.952 $\pm 0.010$ | 0.584 $\pm 0.002$ | 0.948 $\pm 0.006$ |
| w/o zero conv | 0.929 $\pm 0.002$ | 1.619 $\pm 0.009$ | 1.203 $\pm 0.009$ | 1.975 $\pm 0.018$ | 0.214 $\pm 0.001$ | 0.359 $\pm 0.004$ | 0.291 $\pm 0.001$ | 0.469 $\pm 0.007$ |
| GeoAda | **0.891** $\pm 0.003$ | **1.533** $\pm 0.008$ | **1.060** $\pm 0.003$ | **1.852** $\pm 0.012$ | **0.191** $\pm 0.000$ | **0.319** $\pm 0.002$ | **0.240** $\pm 0.002$ | **0.394** $\pm 0.005$ |

**Table 23:** Ablation study of Short-term prediction on **running** from the CMU Mocap dataset.

| scenarios | running | | | | pretrain | | | |
|---|---|---|---|---|---|---|---|---|
| millisecond (ms) | 80 | 160 | 320 | 400 | 80 | 160 | 320 | 400 |
| w/o trainable copy | 44.52 $\pm 0.48$ | 76.98 $\pm 1.20$ | 134.91 $\pm 2.04$ | 159.75 $\pm 2.45$ | 198.16 $\pm 2.97$ | 139.24 $\pm 0.23$ | 270.04 $\pm 0.56$ | 314.89 $\pm 0.68$ |
| w/o zero conv | 19.07 $\pm 0.37$ | 34.25 $\pm 0.82$ | 51.75 $\pm 1.39$ | 55.74 $\pm 1.53$ | nan | nan | nan | nan |
| GeoAda | **18.70** $\pm 0.37$ | **33.56** $\pm 0.25$ | **50.26** $\pm 0.42$ | **55.54** $\pm 0.36$ | **7.972** $\pm 0.02$ | **16.91** $\pm 0.05$ | **40.06** $\pm 0.42$ | **52.61** $\pm 0.07$ |

**Table 24:** Ablation study of Short-term prediction on **walking** from the CMU Mocap dataset.

| scenarios | walking | | | | pretrain | | | |
|---|---|---|---|---|---|---|---|---|
| millisecond (ms) | 80 | 160 | 320 | 400 | 80 | 160 | 320 | 400 |
| w/o trainable copy | 23.77 $\pm 0.15$ | 43.43 $\pm 0.31$ | 84.79 $\pm 0.77$ | 105.08 $\pm 1.28$ | nan | nan | nan | nan |
| w/o zero conv | 12.62 $\pm 0.07$ | 20.67 $\pm 0.20$ | 36.75 $\pm 0.52$ | 44.69 $\pm 0.45$ | 36.18 $\pm 0.12$ | 58.18 $\pm 0.08$ | 102.06 $\pm 0.05$ | 123.82 $\pm 0.04$ |
| GeoAda | **8.92** $\pm 1.02$ | **13.82** $\pm 1.26$ | **22.99** $\pm 1.30$ | **26.68** $\pm 1.31$ | **7.932** $\pm 0.03$ | **16.90** $\pm 0.04$ | **38.96** $\pm 0.47$ | **52.54** $\pm 0.10$ |

**Table 25:** Ablation study of long-term prediction on **running, walking** from the CMU Mocap dataset.

| scenarios | running | | pretrain | | walking | | pretrain | |
|---|---|---|---|---|---|---|---|---|
| millisecond (ms) | 560 | 1000 | 560 | 1000 | 560 | 1000 | 560 | 1000 |
| w/o trainable copy | 198.16 $\pm 2.97$ | 228.41 $\pm 2.96$ | 365.41 $\pm 0.63$ | 374.74 $\pm 0.48$ | 143.35 $\pm 2.26$ | 214.78 $\pm 2.82$ | nan | nan |
| w/o zero conv | 64.69 $\pm 0.97$ | 74.64 $\pm 0.66$ | nan | nan | 60.28 $\pm 1.07$ | 93.99 $\pm 1.41$ | 165.86 $\pm 0.15$ | 249.90 $\pm 0.38$ |
| GeoAda | **60.88** $\pm 0.82$ | **70.22** $\pm 2.02$ | 77.22 $\pm 0.45$ | **130.17** $\pm 0.26$ | **34.52** $\pm 2.26$ | **50.49** $\pm 0.33$ | 78.12 $\pm 0.49$ | **129.97** $\pm 0.30$ |

## 10.4 Standard Deviations

We have already provided the standard deviations in App. 10.2.

## 11 Discussion

**Limitation** While GeoAda demonstrates strong empirical performance and theoretical grounding in preserving SE(3)-equivariance during fine-tuning, several limitations remain: The effectiveness of GeoAda hinges on the design of coupling and decoupling operators for control injection. While theoretically sound, these handcrafted designs may not generalize well to control signals with high-dimensional or structured semantics. Moreover, although GeoAda is validated across multiple domains (e.g., particles, molecules, human motion), the evaluations are limited to medium-scale datasets and relatively small models. Assessing its scalability to larger systems—such as full proteins or macromolecular assemblies—remains an important direction for future work.

