# OpenReview forum: "GeoAda: Efficiently Finetune Geometric Diffusion Models with Equivariant Adapters"
_NeurIPS.cc/2025/Conference — NeurIPS 2025 poster_

### Official Review · Reviewer_CvDD · 2025-06-30

**Clarity:** 3
**Significance:** 2
**Originality:** 2
**Rating:** 5
**Confidence:** 3

**Summary:**

This paper introduces a framework for parameter-efficient finetuning of geometric generative models for downstream tasks that require geometric controls. It proposes an equivariant adapter formulation that is instantiated for 3 types of control inputs: global type control, subgraph control and frame control. A wide range of experiments on particle dynamics simulation, molecule dynamics, pose estimation and docking show favorable results compared to other finetuning approaches.

**Questions:**

1. Can the control types be combined in multi-objective settings where both a global graph property and a subgraph is to be optimized / conditioned?
2. Why did the authors not include recent baselines such as [2], [3], [4] in the experimental evaluation for section 5.3?
3. Can the authors discuss why finetuning for a specific task is favorable compared to training-free conditioning methods such as e.g. (classifier) guidance (as e.g. discussed in [3])?


[2] Guan J, Zhou X, Yang Y, Bao Y, Peng J, Ma J, Liu Q, Wang L, Gu Q. DecompDiff: diffusion models with decomposed priors for structure-based drug design. arXiv preprint arXiv:2403.07902. 2024 Feb 26.

[3] Ayadi, Sirine, et al. "Unified guidance for geometry-conditioned molecular generation." Advances in Neural Information Processing Systems 37 (2024): 138891-138924.

[4] Huang Z, Yang L, Zhou X, Zhang Z, Zhang W, Zheng X, Chen J, Wang Y, Cui B, Yang W. Protein-ligand interaction prior for binding-aware 3d molecule diffusion models. InThe Twelfth International Conference on Learning Representations 2024.

**Ethical Concerns:**

["NO or VERY MINOR ethics concerns only"]

**Final Justification:**

My remaining concerns have been addressed in the rebuttal and I can recommend acceptance. The concerns of other reviewers have also been sufficiently discussed.

**Limitations:**

yes

**Paper Formatting Concerns:**

None.

**Quality:**

2

**Strengths And Weaknesses:**

## Strengths
- the paper is well-motivated and the framework is introduced concisely and clearly.
- the control types are well-presented and cover the most important control inputs in the geometric domain
- the performance of the presented approach compared to other fine-tuning baselines is favorable

## Weaknesses
- the presentation of the contributions is clear and helpful, but is repeated several times throughout the paper. Paragraph 3 and 4 of the Introduction are rather similar, it would help the presentation if the content of those paragraphs is separated more clearly. The final paragraph of Chapter 4 only restates what is mentioned in the introduction and could be significantly shortened.
- while the experimental results look promising and speak for the approach, the presentation of the results needs to be improved. Descriptions for Fig 2, Fig 3, Table 2 do not sufficiently explain the contents of the figures/tables. The background specifically for Tables 6 and 7 is missing; the metrics used in these experiments need introduction/explanation and the overall motivation for these evaluations should be explicitly stated. Highlighting in Table 6 and 7 is missing.
- the related work for geometric diffusion models is sparse and can be extended to give a more complete overview over the literature. EDM ([12] in paper) is mentioned in the introduction but not presented in the related work, neither are more recent works in this line (see [1]). The authors could also consider presenting the relevant baselines from the experimental section (e.g. 5.3 TargetDiff, etc.) here.
- (minor) typos: 121 "are present"? Table 1 "dataset.", 267 'task."', 312 "use and the"

[1] Wang L, Song C, Liu Z, Rong Y, Liu Q, Wu S. Diffusion Models for Molecules: A Survey of Methods and Tasks. arXiv preprint arXiv:2502.09511. 2025 Feb 13.

---

> ### Author Rebuttal · Authors · 2025-07-31
>
> ### ` W1. Rewrite`
> We sincerely thank the reviewer for recognizing the contributions of our work and for the constructive feedback on the writing. We will revise the paper to improve its overall flow and conciseness, and will significantly shorten the final paragraph of Section 4 to avoid redundancy.
> ### ` W2.Table/Figure’s Descriptions`
> We sincerely thank the reviewer for their positive feedback on our experimental results and the effectiveness of our method. We also apologize for the lack of detail in the current figure and table presentations. In the revised version, we will update their captions as follows to provide a clearer and more informative description:
> ```
> Figure2. Overall framework of GeoAda. A control signal C\mathcal{C}C is injected into the noised graph Gτ\mathcal{G}_\tauGτ​ and processed by an equivariant adapter block. The adapter output is added to the frozen denoiser and repeated B times to produce the final symmetry-preserving output.
> ```
> ```
> Figure3.  Visualization results of our GeoAda on Malonaldehyde and Naphthalene from the MD17 dataset.
> ```
> ```
> Table2. Comparisons for Molecular Dynamics prediction on MD17 dataset (all results reported by ×10−1 ). The best results are highlighted in bold. Results averaged over 5 runs. “NaN” denotes generation collapse due to numerical instability, typically observed in baseline models after fine-tuning on the original task.
> ```
> We will also  include the following concise explanation of “Marg,” “Class,” and “Pred” in Section 5.1 to improve readability and facilitate interpretation of the results:
> ```
> The Marginal score measures statistical alignment by computing the mean absolute error (MAE) between binned distributions of model-generated and ground-truth coordinates (or bond lengths for MD17). The Classification score is the cross-entropy of a binary classifier trained to distinguish generated trajectories from real ones, offering insight into sample realism. The Prediction score measures the mean squared error (MSE) of a sequence model trained on generated data and tested on real trajectories, reflecting the utility of generated samples for downstream prediction.  For more detailed metric definitions, please refer to Appendix 8.5.
> ```
> For Tables 6 and 7, we mainly follow the evaluation metrics used in TargetDiff. We apologize for not elaborating on them in the main text due to space limitations. In the revised version, we will include detailed descriptions in the appendix.
>
> ### ` W3. Related work`
> We thank the reviewer for this feedback. In the camera-ready version, we will expand the related work section to provide more comprehensive coverage of the literature. For the works you mentioned, we will include descriptions such as: "EDM introduced an SE(3)-equivariant framework for 3D molecule generation that significantly improved sample quality" and "TargetDiff further extended these models to structure-based drug design by generating molecules conditioned on protein targets through an SE(3)-equivariant processor."
>
> ### ` W4. Typo `
> Thank you for your careful review and for pointing out these typos. We will correct the two errors you mentioned in Lines121& 267 & 312 in the camera-ready version. We will also thoroughly check the rest of the manuscript to ensure no similar issues remain.
>
> ### ` Q1.Multiple Adapters`
> Yes, GeoAda is able to combine multiple adapters which are separately trained by composing the corresponding scores $\mathbf{s}_\theta$ produced by the adapters together during sampling. To demonstrate such potential, we provide an additional experiment on motion capture dataset by separately training two adapters that correspond to frame control (for additional 5 frames) and global type control (following our setup in Sec 5.2), respectively. The model successfully generates prolonged simulation on the new motion (running) with results of 19.10, 33.63, 50.38, 55.92,61.13,71.34 on 80 160 320 400 560 1000ms, showcasing its strong generalization towards composing different controls.
>
> ### ` Q2.Recent Baseline`
> Thank you for raising this. These are highly relevant and timely papers that appeared concurrently with our work. Based on the results reported in these two papers, we summarize the key numbers as follows:
>
> Table 5: Summary of binding affinity and molecular properties of reference molecules and molecules generated by GeoAda and baselines.
>
> | Methods | Vina Score (↓) | Vina Min (↓) | Vina Dock (↓) | High Affinity (↑) | QED (↑) | SA (↑) | Diversity (↑) |
> | ---------------- | ----------------- | ----------------- | ----------------- | ----------------- | --------------- | --------------- | --------------- |
> | | Avg. / Med. | Avg. / Med. | Avg. / Med. | Avg. / Med. | Avg. / Med. | Avg. / Med. | Avg. / Med. |
> | liGAN \[22] | - / - | - / - | -6.33 / -6.20 | 21.1% / 11.1% | 0.39 / 0.39 | 0.59 / 0.57 | 0.66 / 0.67 |
> | GraphBP \[16] | - / - | - / - | -4.80 / -4.70 | 14.2% / 6.7% | 0.43 / 0.45 | 0.49 / 0.48 | 0.79 / 0.78|
> | AR \[18] | -5.75 / -5.64 | -6.18 / -5.88 | -6.75 / -6.62 | 37.9% / 31.0% | 0.51 / 0.50 | 0.63 / 0.63 | 0.70 / 0.70 |
> | Pocket2Mol \[21] | -5.14 / -4.70 | -6.42 / -5.82 | -7.15 / -6.79 | 48.4% / 51.0% | 0.56 / 0.57 | 0.74 / 0.75 | 0.69 / 0.71 |
> | TargetDiff \[8] | -5.47 / -6.30 | -6.64 / -6.83| -7.80 / -7.91 | 58.1% / 59.1% | 0.48 / 0.48 | 0.58 / 0.58 | 0.72 / 0.71 |
> |DecompDiff| -5.67 -6.04| -7.04 -7.09| -8.39 -8.43| 64.4% 71.0% |0.45 0.43 |0.61 0.60| 0.68 0.68 |
> | IPDIFF |-6.42 -7.01 |-7.45 -7.48 |-8.57 -8.51 |69.5% 75.5% |0.52 0.53| 0.61 0.59 |0.74 0.73|
> | GeoAda (qm9) | -5.54 / -6.31| -6.46 / -6.46 | -7.62 / -7.64 | 57.4% / 58.2% | 0.49 / 0.51 | 0.58 / 0.58 | 0.74 / **0.75** |
> | GeoAda (Geom) | -5.54 / -6.01| -6.68 / -6.32 | -7.64 / -7.71 | 58.3% / 59.3% | 0.48 / 0.50 | 0.58 / 0.58 | 0.76 / 0.75 |
> | Reference | -6.36 / -6.41 | -6.71 / -6.49 | -7.45 / -7.26 | - / - | 0.48 / 0.47 | 0.73 / 0.74 | - / - |
>
> Table 6: Jensen-Shannon divergence of bond distance distributions between reference and generated molecules.
>
> | Bond | liGAN | AR    | Pocket2Mol | TargetDiff | DecompDiff|IPDIFF | GeoAda (qm9) | GeoAda (Geom) |
> | ---- | ----- | ----- | ---------- | ---------- | ------------ | ------------- | ------------ | ------------- |
> | C–C  | 0.601 | 0.609 | 0.496      | 0.369  |0.359| 0.386    | 0.243   | 0.269         |
> | C=C  | 0.665 | 0.620 | 0.561      | 0.505   |  0.537 | 0.245  | 0.377    | 0.393         |
> | C–N  | 0.634 | 0.474 | 0.416      | 0.363   |   0.344  |0.298  | 0.363    | 0.396         |
> | C=N  | 0.749 | 0.635 | 0.629      | 0.550   | 0.584  |0.238   | 0.300    | 0.299    |
> | C–O  | 0.656 | 0.492 | 0.454      | 0.421 |  0.376  |0.366    | 0.418   | 0.428         |
> | C=O  | 0.661 | 0.558 | 0.516      | 0.461    | 0.374  |0.353   | 0.279    | 0.257         |
> | C\:C | 0.497 | 0.451 | 0.416      | 0.263     | 0.251 | 0.169  | 0.305    | 0.335         |
> | C\:N | 0.638 | 0.552 | 0.487      | 0.235  | 0.269  |0.128| 0.297        | 0.330         |
>
> TargetDiff trains molecular generation simply conditioned on protein, and there is future work, such as IP-Diff that  further adds protein and molecule interaction as additional conditions. Our work builds on the TargetDiff pipeline, taking off the protein condition during pretraining and adding trainable copy layers for Finetuning, which is the key reason why we mainly compared our result with TargetDiff and may not outperform the latest SOTA methods in every dimension.
>
> But GeoAda consistently improves upon TargetDiff, demonstrating the effectiveness and generality. Moreover, GeoAda is modular by design and can be easily applied to other backbones, making it broadly applicable across future geometric diffusion models.
>
>
> ### `Q3. FT vs Unified guidance`
> We appreciate the reviewer for bringing this up — this is an insightful comparison and a key point that reinforces the motivation of our method.
> While methods like classifier guidance can offer strong control, they typically require training a separate predictor for each property and rely on adversarial gradients during sampling.This restricts their applicability to tasks lacking reliable property-specific classifiers and makes the generation process more prone to instability introduced by adversarial optimization.
>
> In contrast, our fine-tuning approach employs lightweight, easily pluggable adapters without requiring additional predictors. These adapters also provide implicit regularization, reducing reward hacking risks and ensuring stable control. Moreover, our method generalizes effectively across diverse tasks—including frame, global, and subgraph control—where training-free methods may fall short.
>
> Nonetheless, we acknowledge that both fine-tuning and guidance are complementary and promising directions. The choice between them may depend on the specific requirements of the target task.
>
> ## Summary
> We hope our responses sufficiently address your concerns regarding: (1) Writing Redundancy and Clarity, (2) Table and Figure Presentation, (3) Metric Definitions, (4) Related Work Coverage, (5) Minor Typos, (6) Multi Control Composition, (7) Inclusion of Recent Baselines, and (8) Fine-Tuning vs. Training-Free Guidance.
>
> We sincerely thank the reviewer for recognizing the clarity of our framework, the relevance of our control settings, and the strong empirical performance across tasks. We truly appreciate your reconsideration of our work in light of our responses. Thank you again for your valuable and thoughtful feedback.

---

> > ### Comment · Reviewer_CvDD · 2025-08-04
> >
> > Thank you for the thoughtful rebuttal. My concerns have been addressed and I will raise my score from 4 to 5 and recommend acceptance.

---

> > > ### Author Response · Authors · 2025-08-05
> > >
> > > Dear Reviewer CvDD,
> > >
> > > Thank you for your thoughtful comments. We’re glad to hear that all your questions have been addressed. We truly appreciate your recognition of our contributions and your willingness to support the acceptance of our work.
> > >
> > > Best regards,
> > >
> > > The Authors

---

### Official Review · Reviewer_TDhS · 2025-07-01

**Clarity:** 2
**Significance:** 2
**Originality:** 2
**Rating:** 4
**Confidence:** 4

**Summary:**

This paper presents GeoAda, an SE(3)-equivariant adapter framework for efficient fine-tuning of geometric diffusion models under diverse geometric conditions. GeoAda integrates lightweight, plug-in adapters into a frozen pretrained denoiser, using coupling/decoupling operators, selectively trainable layers, and zero-initialized equivariant convolutions for stable optimization. It preserves SE(3)-equivariance and achieves strong transfer performance across tasks like dynamics simulation and molecule generation, outperforming or matching full fine-tuning while reducing overfitting and forgetting.

**Questions:**

1. In Table 1, the metrics "Marg, Class, and Pred" should be introduced more clearly in the main text rather than only in the appendix. Providing brief descriptions upfront would improve clarity and help readers better interpret the results.
2. The presence of numerous "NaN" results in the experiments raises concerns. It would be helpful to clarify the underlying reasons for these failures, particularly in relation to the specific baselines, to provide a more complete and transparent evaluation.
3. What specific pretrained model is used for each task? This information seems to be missing—could the authors clarify it to improve transparency and reproducibility?
4. In the "task" row of the table, it’s recommended to replace "FT" with "Downstream" to avoid potential confusion and improve clarity.
5. Why doesn’t the Subgraph Control setting include results on the original (pretrain) task? Including them would help assess whether control fine-tuning preserves or compromises performance on the base task.

**Note**: I will raise my score if the authors solve my concerns.

**Ethical Concerns:**

["NO or VERY MINOR ethics concerns only"]

**Final Justification:**

The authors have addressed my concerns regarding the ablation experiments. As a result, I have increased my recommendation score from 3 to 4, and my confidence level from 2 to 4.

**Limitations:**

As mentioned in the **Weakness** part, the method may fail in out-of-distribution (OOD) settings, which are critical in real-world applications. Could the authors explore ways to improve the approach under OOD scenarios? Addressing this limitation would significantly strengthen the paper's contribution.

**Quality:**

3

**Strengths And Weaknesses:**

### Strengths
1. The inability of existing geometric diffusion models to efficiently adapt to new geometric control tasks is convincingly argued, addressing a key pain point for real-world applications.
2. The method is well-motivated, with sound theoretical backing. The authors provide a proof  that their adapter construction retains SE(3)-equivariance, an essential property for geometric models.
3. GeoAda offers an architecture-agnostic mechanism to inject control into geometric diffusion models using equivariant adapters, drawing inspiration from image diffusion adapters but thoughtfully extending them to the geometric (3D) and equivariant setting.
4. Experiments span a diverse set of domains. The results consistently show that GeoAda matches or outperforms strong baselines (such as full fine-tuning, prompt and head-only tuning) across frame, global, and subgraph control tasks. Notably, in Table 5 and Table 6, key molecular property metrics and distributional metrics demonstrate superior or competitive performance.

### Weaknesses
1. While the method claims that choosing which layers to copy/introduce adapters into can matter for effectiveness (Section 4.3), there is insufficient discussion in the main paper on how sensitive results are to these architectural decisions. More quantitative results  regarding how the choice of adapter positions, number of inserted blocks, and their impact on expressivity/regularization would strengthen the argument.
2. The paper emphasizes parameter efficiency but provides only minimal quantitative analysis of actual wall-clock training time, memory usage, or scalability to very large models. A more detailed treatment—with explicit profiling, speedup ratios, or resource consumption figures for different settings—would be informative to practitioners.
3. The experimental section covers several domains, but does not fully assess the adapter’s robustness to transfer across out-of-distribution (OOD) controls or tasks, e.g., how performance degrades if the new task features significantly different structural characteristics than seen during pretraining.
4.  While Figures 1 and 2 effectively clarify the architecture, more qualitative or structural analysis of what the adapters actually learn (e.g., via feature attribution or intermediate representation visualization) could provide deeper insights.

---

> ### Author Rebuttal · Authors · 2025-07-31
>
> Thank you for your detailed review and valuable feedback. We have addressed your concerns below and hope these clarifications will assist you in re-evaluating and update the score:
>
> ### ` W1. Sensitive Results`
> Thank you for pointing this out — this is indeed an important aspect. To better quantify the impact of architectural choices, we’ve conducted ablation study on the number of trainable copy layers, reported in Section 5.4 and Appendix 9.3.1 (Table 20). The results show that increasing the number of trainable layers generally improves performance, but also introduces additional parameters and computational cost, highlighting a trade-off between accuracy and efficiency.
>
> We further ablated the architectural design of our adapters in Section 5.4 and Appendix 9.3.2. Specifically, we evaluated two variants to assess the role of the equivariant zero convolution: (1) replacing it with standard Gaussian-initialized convolutions, and (2) replacing each trainable copy block with a single convolution layer. Both modifications resulted in noticeable performance degradation, emphasizing the importance of zero initialization and the proposed structural design for stable and effective fine-tuning.
>
> We apologize for not presenting these results more clearly in the main paper due to space limitations. In the revised version, we will include key tables and results in the main text to improve clarity and visibility.
>
> ### ` W2. Usage details`
> We apologize for the confusion, and we completely agree that parameter analysis is important. As mentioned in Section 5.4, we report the number of tunable parameters for different tuning strategies in Appendix 9.3, Table 21. Specifically, Full Fine-Tuning involves 1,418,252 tunable parameters (\~5.41 MB), whereas GeoAda only requires 711,791 tunable parameters (~2.72 MB), effectively reducing the parameter count by half. Despite this reduction, GeoAda achieves better or comparable performance on downstream tasks while also preserving performance on the original task,  avoiding issues such as model forgetting or collapse.
> We will make sure to highlight this quantitative analysis more clearly in the main text of the revised version.
>
> ### ` W3. OOD result`
> Evaluating robustness to OOD controls is indeed a valuable aspect. We further evaluate our model on OOD task involving frame control on Charged Particle setting conditioning on 5 frames to predict the next 20 frames.
>
> |                | OOD task 5          |                  | Downstream task 15 |                    |      Pretrain task 10 |                  |
> |:----------------:|:----------------------:|:------------------:|:------------------------:|:------------------:|:--------------------------:|:------------------:|
> |                | ADE                  | FDE              | ADE                    | FDE              | ADE                      | FDE              |
> | **Base**       | 12.1955±0.089*   | 20.8427±0.247| 11.826±0.133      | 20.395±0.249 | 1.177±0.018         | 2.815±0.037 |
> | **SFT**        |   338.02   ± 4.36    |   340.74   ± 4.42        | 1.106±0.007            | 2.590±0.040      | 5.998±0.041              | 11.75±0.107      |
> | **GeoAda**   | 1.0771±0.005        | 2.3864±0.022     | 1.105±0.012            | 2.621±0.033      | 1.175±0.033              | 2.806±0.033      |
>
> These results highlight the strong OOD generalization and downstream performance of GeoAda.
>
> ### ` W4. Feature`
> Thank you for this suggestion. To provide deeper insight into the behavior of the adapters, we generated visualizations comparing the outputs of the original (frozen) network, the trainable copy, and the ground truth during inference. These comparisons help illustrate how the adapter modifies the generation trajectory and captures control-relevant structures. Due to the rebuttal policy restricting figures, we are unable to include these results here, but we will incorporate them in the camera-ready version to support a more comprehensive understanding of the model's effect.
>
> ### ` Q1. Metrics Descriptions in Main Paper`
> We fully agree that these metrics should be better introduced in the main text to offer more background for the reader. In the revised version, we will include the following concise explanation of “Marg,” “Class,” and “Pred” in Section 5.1 to improve readability and facilitate interpretation of the results:
> ```
> The Marginal score measures statistical alignment by computing the mean absolute error (MAE) between binned distributions of model-generated and ground-truth coordinates (or bond lengths for MD17). The Classification score is the cross-entropy of a binary classifier trained to distinguish generated trajectories from real ones, offering insight into sample realism. The Prediction score measures the mean squared error (MSE) of a sequence model trained on generated data and tested on real trajectories, reflecting the utility of generated samples for downstream prediction.  For more detailed metric definitions, please refer to Appendix 8.5.
> ```
> ### ` Q2. NaN `
> “NaN” indicates that the model experienced generation collapse and failed to produce valid outputs due to numerical instability. This issue was observed in certain baseline models after fine-tuning, particularly when evaluated on the original task, where they were unable to generate numerically valid samples. This further highlights the necessity of our method, which preserves stability and avoids such collapse.
> We sincerely thank the reviewer for pointing out the lack of explanation for these two cases. In the revised version, we will update the caption of Table 1 as follows to provide a clearer description:
> ```
> Table 1. Comparisons on the CHARGED PARTICLES dataset (all results reported as ×10⁻¹). (↑)/(↓) indicates whether a higher/lower value is preferred. “NaN” denotes generation collapse due to numerical instability, typically observed in baseline models after fine-tuning on the original task. “–” indicates that the baseline Prompt-Tem  requires explicit conditioning and cannot be applied when no conditioning frame is given.
> ```
>
> ### ` Q3. Backbone & Pretrained Model`
> We sincerely apologize for the lack of clarity regarding experimental details in the main paper and will revise the manuscript to include a more thorough description.
>
> For both frame control and global control settings, we follow the same setup as GeoTDM [1]. The backbone model used in our method is EGTN, which is also employed for all baselines to ensure fair comparison. The original base model consists of six EGTN layers. We insert three trainable copies at layers 1, 3, and 5. That is, the model architecture becomes:
> ```
> input → layer1_origin + layer1_copy → layer2_origin → layer3_origin + layer3_copy
> → layer4_origin → layer5_origin + layer5_copy → layer6_origin → output.
> ```
> For subgraph control, the base model is following the setting of TargetDiff [2], which has nine layers in total. We apply the same strategy by inserting trainable copies at layers 1, 3, 5, and 7.
>
> [1] Geometric trajectory diffusion models.
>
> [2] 3d equivariant diffusion for target-aware molecule generation and affinity prediction.
>
> ### ` Q4. Change FT with  Downstream`
> Thank you for this valuable suggestion to improve clarity! We will replace “FT” with “Downstream” in all tables in the revised version.
> ### ` Q5. Subgraph Control on Pretrain`
> On the subgraph control task, all baseline methods are trained from scratch on the downstream CrossDocked dataset with protein pocket conditions. In contrast, our method is first pretrained on unconditional datasets (QM9 and GEOM-Drugs) and then fine-tuned on the downstream task, demonstrating GeoAda’s effectiveness and generalizability. Therefore, we do not report baseline performance on QM9 or GEOM-Drugs. However, this does not affect our conclusions, as our method is designed to preserve performance on the pretraining dataset while enabling effective adaptation to new tasks.
>
> ## Summary
> We hope our answers sufficiently address your concerns regarding: (1) Sensitive study Results, (2) Parameter Efficiency, (3) OOD Robustness, (4) Adapter Behavior Visualization, (5) Metric Descriptions, (6) “NaN” Clarification, (7) Pretrained Backbone and Architecture, (8) Table Label Clarity, and (9) Subgraph Control results on pretrain.
>
> We also sincerely thank the reviewer for the positive feedback on the motivation, theoretical soundness, and practical relevance of our method, as well as the recognition of our strong and diverse experimental results. We truly appreciate your reconsideration of our work in light of our responses. Thank you once again for your valuable insights.

---

> > ### Comment · Reviewer_TDhS · 2025-08-03
> >
> > I appreciate the authors' thorough responses. My questions have been addressed, and I am now confident in the quality of the work. I will increase my evaluation score accordingly. If the paper is accepted, the authors are supposed to add the corresponding modifications to the camera-ready version.

---

> > > ### Author Response · Authors · 2025-08-05
> > >
> > > Dear Reviewer TDhS
> > >
> > > We sincerely thank you for your encouraging feedback on our rebuttal! We will definitely incorporate your valuable suggestions in our revised version. Thanks again for your valuable time and efforts in reviewing our paper!
> > >
> > > Warm regards,
> > >
> > > Authors

---

### Official Review · Reviewer_iYKL · 2025-07-01

**Clarity:** 3
**Significance:** 3
**Originality:** 2
**Rating:** 4
**Confidence:** 4

**Summary:**

This paper introduces GeoAda, a framework for adapting pre-trained geometric diffusion models for conditional generation by learning adapter blocks on top of the frozen pre-trained model. GeoAda is conceptually similar to the ControlNet architecture [1], which was developed for controlling text-to-image diffusion models, and extends it to the geometric domain by designing SE(3)-equivariant adapter blocks. GeoAda covers different control types and shows promising experimental results across a variety of tasks.

**Questions:**

1. In subgraph control, can the control signal be part of the graph itself (and not a separate graph as in the pocket-conditioned example)? For example, can you condition on a scaffold of a molecule and complete the rest? If yes, how does this fit into the proposed framework?
2. In global type control, how is the MLP in the coupling operator $f$ trained? Is it jointly trained with the rest of the model? Also, line 149 says "linear" transformation, which is not correct if an MLP is used (assuming it has some non-linearities).
3. How exactly do you select the trainable copy for fine-tuning? How is "selecting the first layer for every K consecutive layers from the pretrained model" actually implemented?
4. Can you combine multiple adapters, as in ControlNet? Can you also combine them if they are independently trained? For example, one might want to condition on a protein pocket and a target property at the same time.
5. Equation (6) uses $\phi_h \cdot h_i$, where $\phi_h \in \mathbb{R}^H$, which seems inconsistent. Should $\phi_h$ be a scalar, or should the multiplication be an element-wise product?
6. In the results tables, what does "nan" refer to? And in Table 1, what does "-" refer to?

**References**

[1] Zhang, Lvmin, Anyi Rao, and Maneesh Agrawala. "Adding conditional control to text-to-image diffusion models." Proceedings of the IEEE/CVF international conference on computer vision. 2023.

**Ethical Concerns:**

["NO or VERY MINOR ethics concerns only"]

**Final Justification:**

The authors clearly answered my questions. However, while the paper nicely adapts the ControlNet architecture to the geometric domain, the core idea is similar. Therefore, I maintain my score of 4.

**Limitations:**

yes

**Quality:**

2

**Strengths And Weaknesses:**

**Strengths**

* The problem of adapting pre-trained geometric diffusion models to different downstream tasks is interesting and has many real-world applications.
* The paper is well-written with a clear structure and presentation.
* The proposed method is well-motivated and technically sound.
* The experiments cover a wide range of applications and show the good performance of the proposed method.

**Weaknesses**

* The core idea of the paper is taken from the ControlNet architecture [1], which limits the novelty aspect of the paper. However, the paper successfully adapts it to the geometric domain by designing adequate coupling and decoupling operators and ensuring the adapter block is equivariant. The authors should clearly highlight their own contributions and disentangle them from prior work.
* Some details about the experimental setups are not clear, which makes it hard to interpret the results. For example, it is not clear which model architecture is used for each task, and how the trainable copy is selected from the pre-trained model. Also, the human pose estimation experiments are not clear (what is the input/output of the model?, how do you choose the joints?)

---

> ### Author Rebuttal · Authors · 2025-07-31
>
> We sincerely appreciate your positive assessment and valuable suggestions on improving our paper. Regarding your questions, we detailed our responses below.
> ### `W1. Novelty beyond ControlNet`
> We agree that ControlNet is a key inspiration for our work. However, as the reviewer also mentioned, our primary contribution and novelty lie in successfully adapting and extending this paradigm to the geometric deep learning domain which is non-trivial. Our core technical innovations include:
> * Designing domain-specific coupling/decoupling operators that can inject diverse geometric conditions (e.g., global, subgraphs, Frame control) into a pre-trained model.
> * Ensuring the adapter architecture is equivariant, which is a fundamental requirement for most geometric tasks and not an aspect of the original ControlNet.
> * Demonstrating the versatility of this approach across varied and important geometric tasks like particle /molecular dynamics, human motion, and molecular generation.
>
> We will restructure the introduction and related work sections to more explicitly disentangle our contributions from ControlNet and highlight the unique challenges and solutions for geometric data.
> ### `W2. Experimental Details`
> We sincerely apologize for the lack of clarity regarding experimental details in the main paper and will revise the manuscript to include a more thorough description.
>
> **Model Architectures & Trainable Copy Strategy:**
>
>  For both trajectory control and global control settings, we follow the same setup as GeoTDM [1]. The backbone model used in our method is EGTN, which is also employed for all baselines to ensure fair comparison. The original base model consists of six EGTN layers. We insert three trainable copies at layers 1, 3, and 5. That is, the model architecture becomes:
> ```
> input → layer1_origin + layer1_copy → layer2_origin → layer3_origin + layer3_copy
> → layer4_origin → layer5_origin + layer5_copy → layer6_origin → output.
> ```
> For subgraph control, the base model is following the setting of TargetDiff [2], which has nine layers in total. We apply the same strategy by inserting trainable copies at layers 1, 3, 5, and 7.
>
> We further perform an ablation study on the number of trainable copy layers in Section 5.4 and Appendix 9.3.1 (Table 20). The results show that increasing the number of trainable layers generally improves performance, but also leads to higher parameter count and computational cost, revealing a trade-off between accuracy and efficiency.
>
> **Human Motion**
>
> The Human Mocap dataset is a widely adopted benchmark for human pose prediction. Due to space limitations, the detailed dataset splits and experimental setup for our human motion experiments are provided in Appendix 8.3.1. For the input/output formats and joint selection, we closely follow the configuration used in MSR-GCN [3], taking in 10 Frames and predicting the following 25 frames.
>
> We sincerely thank the reviewer for pointing out these sources of confusion. In the revised version of the paper, we will explicitly reference both this appendix section and the relevant prior work in the main text to enhance clarity and improve accessibility for readers.
>
> [1] Geometric trajectory diffusion models.
>
> [2] 3d equivariant diffusion for target-aware molecule generation and affinity prediction.
>
> [3] MSR-GCN: Multi-Scale Residual Graph Convolution Networks for Human Motion Prediction
>
> ### ` Q1. Subgraph Control`
> Thank you for raising this interesting question. In the case that the subgraph control $ \mathcal{C} $ is part of the graph $ \mathcal{G} $ itself, we can instead implement the coupling operator f as an identity function since the original graph is already a supergraph of the subgraph control. For the decoupling operator, we can keep it the same that extracts the nodes that need to optimize from the output of the model and discards the nodes in the subgraph control $ \mathcal{C}$.
>
> ### ` Q2. Global type Control`
> Yes, the coupling function $f$ is trained together with the rest of the model. In line 150 we introduce $ f $ as an MLP as a general case, while we practically find that a simple linear layer already works well enough for $ f $ thus having kept such a design. We will clarify this point in the final version.
> ### ` Q3. Selection of Trainable copies`
> As mentioned in W2,
> ```
> input → layer1_origin + layer1_copy → layer2_origin → layer3_origin + layer3_copy
> → layer4_origin → layer5_origin + layer5_copy → layer6_origin → output.
> ```
> is an example of selecting the first layer of every K=2 layers. We will incorporate the corresponding clarifications into the main text in the revised version.
>
> ### ` Q4. multiple adapters`
> Yes, GeoAda is able to combine multiple adapters which are separately trained by composing the corresponding scores $\mathbf{s}_\theta$ produced by the adapters together during sampling.
>
> To demonstrate such potential, we provide an additional experiment on motion capture dataset by separately training two adapters that correspond to frame control (for additional 5 frames) and global type control (following our setup in Sec 5.2), respectively. The model successfully generates prolonged simulation on the new motion (running) with results of 19.10, 33.63, 50.38, 55.92, 61.13, 71.34 on 80 160 320 400 560 1000ms, showcasing its strong generalization towards composing different controls.
> ### `Q5. Equation (6)`
> Thank you for pointing out the ambiguity in Equation (6). We confirm that $\phi_h$ is a vector, and the use of the dot symbol “$\cdot$” was a typo. The intended operation is element-wise multiplication, and we will correct the symbol to “$\odot$” in the final manuscript to ensure clarity.
> ### ` Q6. nan & –`
> Thank you for pointing out this confusion.
>
> “NaN” indicates that the model experienced generation collapse and failed to produce valid outputs due to numerical instability. This issue was observed in certain baseline models after fine-tuning, particularly when evaluated on the original task, where they were unable to generate numerically valid samples. This further highlights the necessity of our method, which preserves stability and avoids such collapse.
>
> The “–” in Table 1 for Prompt-Tem under the unconditional setting denotes that this baseline is not applicable. Prompt-Tem is specifically designed to match a known conditioning frame (e.g., frame 5 to frame 10), and thus cannot be meaningfully applied to unconditional generation scenarios where no such conditioning exists.
>
> We sincerely thank the reviewer for pointing out the lack of explanation for these two cases. In the revised version, we will update the caption of Table 1 as follows to provide a clearer description:
> ```
> Table 1. Comparisons on the CHARGED PARTICLES dataset (all results reported as ×10⁻¹). (↑)/(↓) indicates whether a higher/lower value is preferred. “NaN” denotes generation collapse due to numerical instability, typically observed in baseline models after fine-tuning on original task. “–” indicates that the baseline Prompt-Tem requires explicit conditioning and cannot be applied when no conditioning frame is given.
> ```
> ## Summary
> We hope our answers sufficiently address your concerns regarding: (1) Novelty Clarification, (2) Experimental Details,(3) Subgraph Control Integration, (4) Global Control MLP Training, (5)Trainable Copy Selection,(6) Multiple Adapter Composition, (7) Equation. 6 Notation, and (8)“NaN” and “–” Clarification.
>
> We also sincerely thank the reviewer for the positive feedback on both the technical soundness of our approach and the breadth of our experimental validation. We truly appreciate your reconsideration of our work in light of our responses. Thank you once again for your valuable insights.

---

> > ### Comment · Reviewer_iYKL · 2025-08-04
> >
> > Thank you for the detailed response! It addressed all my questions. I maintain my recommendation for acceptance.

---

> > > ### Author Response · Authors · 2025-08-05
> > >
> > > Dear Reviewer iYKL
> > >
> > > Thank you for your kind words and for taking the time to review our response. We are glad to hear that you have received answers to all of your queries, and we also appreciate your recognition of our work. Thank you for acknowledging our efforts.
> > >
> > > Warm regards,
> > >
> > > Authors

---

### Official Review · Reviewer_P2xL · 2025-07-03

**Clarity:** 3
**Significance:** 3
**Originality:** 2
**Rating:** 5
**Confidence:** 3

**Summary:**

The authors propose a novel adapter design for the parameter-efficient fine tuning of pre-trained GNN-based SE(3)-equivariant diffusion models. Their framework allows for adding new flexible conditioning signals and are designed for efficient and stable training from the proposed zero initialisation. In addition, the authors ensure that SE(3) equivariance is preserved by their adapter design.

**Questions:**

- It appears that Equation 7 is a bit misleading as the adapter acts on the intermediate activations of the original model $\epsilon_\theta$ not just the final layer. Is there any reason for why this choice of notation is used instead?

**Ethical Concerns:**

["NO or VERY MINOR ethics concerns only"]

**Final Justification:**

I believe the paper is worthy of acceptance and the authors have further addressed the points raised in the review. I will maintain my rating at 5 for acceptance.

**Limitations:**

yes

**Quality:**

3

**Strengths And Weaknesses:**

Strengths:

- Address an important yet underexplored area of fine-tuning geometric diffusion models.
- Nice, simple idea and implementation with compelling results.
- Good range of experiments, ablations and baselines to validate the framework.

Weaknesses:

- The essential idea is not too novel due to prior work on ControlNet.

Typos:

- Line 51: missing a space in "work(GeoAda)"
- Line 138: should be "Such a feature" instead of "Such feature"

---

> ### Author Rebuttal · Authors · 2025-07-31
>
> We sincerely appreciate your positive assessment and valuable suggestions on improving our paper. Regarding your questions, we detailed our responses below.
> ### ` W1. Novelty beyond ControlNet`
>
> We agree that ControlNet is a key inspiration for our work. However, our primary contributions and novelty lie in successfully adapting and extending this paradigm to the geometric deep learning domain which is non-trivial. Our core technical innovations include:
> * Designing domain-specific coupling/decoupling operators that can inject diverse geometric conditions (e.g., global, subgraphs, Frame control) into a pre-trained model.
> * Ensuring the adapter architecture is equivariant, which is a fundamental requirement for most geometric tasks and not an aspect of the original ControlNet.
> * Demonstrating the versatility of this approach across varied and important geometric tasks like particle /molecular dynamics, human motion, and molecular generation.
> We will restructure the introduction and related work sections to more explicitly disentangle our contributions from ControlNet and highlight the unique challenges and solutions for geometric data.
>
> ### ` W1.Typos`
> Thank you for your careful review and for pointing out these typos. We will correct the two errors you mentioned in Lines 51 & 138 in the camera-ready version. We will also thoroughly check the rest of the manuscript to ensure no similar issues remain.
>
> ### `Q1. Notation in Eq. (7)`
> Thank you for this insightful question. You are correct that our adapter acts on intermediate activations, not just the final layer, as shown in Figure 2, where the adapter $s_{\theta', \phi} $ processes the control $ \mathcal{C} $ in addition to the original inputs $ (\mathcal{G}_\tau, \tau) $.
>
> We use the notation in our loss function to express the high-level training objective in a clear and compact way:
> This formula defines the overall goal, training the adapter $ s_{\theta', \phi} $ to produce a corrective signal that is added to the frozen model’s $ \epsilon_\theta $ prediction.
> The layer-by-layer additions in our architecture are the concrete mechanism to achieve this high-level objective.
>
> We acknowledge this distinction could be clearer and will revise Section 3.2  to explicitly connect the objective function with its layer-wise architectural implementation.
>
>
> ## **Summary**
> We hope our answers address your concerns. Thanks again for recognizing our contribution to this important and challenging topic, as well as the effectiveness of our approach and the comprehensiveness of our evaluation.

---

> > ### Comment · Reviewer_P2xL · 2025-08-01
> >
> > Thank you for the response to the points raised in the review. I will maintain my recommendation of acceptance.

---

> > > ### Author Response · Authors · 2025-08-05
> > >
> > > Dear Reviewer P2xL
> > >
> > > Thank you so much for your positive feedback! We sincerely appreciate your valuable time and efforts in reviewing our submission.
> > >
> > > Warm regards,
> > >
> > > Authors

---

### Decision · Program_Chairs · 2025-09-17

**Decision:**

Accept (poster)

**Comment:**

The paper introduces an SE(3)-equivariant adapter framework for parameter-efficient fine-tuning of geometric diffusion models. Reviewers found the work well-motivated, well-written, technically sound, and supported by comprehensive experiments showing competitive performance. While concerns were raised about limited novelty and missing experimental details, these were mostly addressed during the rebuttal, leading to consistent acceptance recommendations. I therefore recommend accepting the paper.